# Molecular characterization and prospective isolation of human fetal cochlear hair cell progenitors

Marta Roccio[1,2], Michael Perny[1,2,3], Megan Ealy[4,5,6], Hans Ruedi Widmer[7], Stefan Heller[4,5] & Pascal Senn[1,2,8]

Sensory hair cells located in the organ of Corti are essential for cochlear mechanosensation. Their loss is irreversible in humans resulting in permanent hearing loss. The development of therapeutic interventions for hearing loss requires fundamental knowledge about similarities and potential differences between animal models and human development as well as the establishment of human cell based-assays. Here we analyze gene and protein expression of the developing human inner ear in a temporal window spanning from week 8 to 12 post conception, when cochlear hair cells become specified. Utilizing surface markers for the cochlear prosensory domain, namely EPCAM and CD271, we purify postmitotic hair cell progenitors that, when placed in culture in three-dimensional organoids, regain proliferative potential and eventually differentiate to hair cell-like cells in vitro. These results provide a foundation for comparative studies with otic cells generated from human pluripotent stem cells and for establishing novel platforms for drug validation.

[1] Laboratory of Inner Ear Research, Department of Biomedical Research, University of Bern, 3008 Bern, Switzerland. [2] Department of Otorhinolaryngology, Head and Neck Surgery, Inselspital Bern, 3008 Bern, Switzerland. [3] Neuroinfection Laboratory, Institute for Infectious Diseases, University of Bern, 3008 Bern, Switzerland. [4] Department of Otolaryngology, Head and Neck Surgery, Stanford University School of Medicine, Stanford, CA 94305, USA. [5] Institute for Stem Cell Biology and Regenerative Medicine, Stanford University School of Medicine, Stanford, CA 94305, USA. [6] Department of Biology, Drury University, Springfield 65802 MO, USA. [7] Department of Neurosurgery, Neurocenter and Regenerative Neuroscience Cluster, Inselspital Bern, Bern 3008, Switzerland. [8] Department of Clinical Neurosciences, Service of Otorhinolaryngology and Head and Neck Surgery, Hôpitaux Universitaires de Genève, Geneva 1211, Switzerland. Correspondence and requests for materials should be addressed to M.R. (email: marta.roccio@dbmr.unibe.ch)

Hearing in humans relies on mechanosensitive hair cells located in the organ of Corti. Hair cells and their surrounding non-sensory supporting cells derive from SOX2+ progenitors within a region of the developing cochlear duct known as the prosensory domain (PSD)[1]. The PSD becomes postmitotic as early as embryonic day E12.5–E13 in mice[2]. Expression of the cell cycle inhibitor p27Kip1, progressing in an apical-to-basal gradient, coincides with cell cycle exit[3]. Hair cells and supporting cells are specified shortly after by coordinated activity of transcription factors, such as Atoh1[4–7], and by Notch-mediated lateral inhibition[8,9], resulting in a mosaic-like pattern of the two cell types[10].

While extensive data are available on gene expression during mouse development, only limited information exists for human cochlear development[11–13]. The first appearance of hair cells within the human cochlear duct has previously been reported during the 12–13th week of development[12]. The earliest occurrence of human otic neuroblasts and the appearance of vestibular hair cells has not been well documented.

Characterization of the fetal PSD could provide a framework for understanding human hair cell development and for comparative studies with the goal of finding ways to initiate hair cell regeneration in the human cochlea. Moreover, gaining information about human hair cell progenitors will offer a blueprint to generate this rare and transient human cell type from more abundant sources such as pluripotent stem cells[14,15].

Here we mapped the expression of well-known otic markers by immunohistochemistry and multiplex qRT–PCR during human inner ear development. We focused on the developmental stages when the human cochlear PSD becomes postmitotic and hair cells start to differentiate; in parallel we characterized the spiral ganglion as well as the vestibular sensory epithelium. Moreover, we have developed an organoid culture method that allows for expansion of human fetal cochlear duct cells upon fluorescence activated cell sorting (FACS), relying on EPCAM expression. We show that a cell population expressing EPCAM and CD271 includes nearly the totality of hair cell progenitors within the human cochlear PSD. Our results provide insights in the development of the human inner ear and provide a method to purify and culture human hair cell progenitors and differentiate them in vitro to hair cell-like cells.

## Results

**The human cochlear prosensory domain**. Cell cycle exit in the murine cochlear PSD begins at the apex of the organ during embryonic day 12 and migrates toward its base during the course of 24 h[2]. An indicator of PSD cell cycle exit is the onset of expression of the cyclin-dependent kinase inhibitor 1B (CDKN1B), also known as p27Kip1[3,16]. We analyzed expression of p27Kip1 in human samples from the eighth week (W8) until W12 of development (Fig. 1a–e). In W8 cochleae, p27Kip1 expression was detectable in cells of the floor of the developing cochlear duct in apical and middle turns, but not at the base (Fig. 1a, b). Reciprocally, and in accordance with an apex-to-base gradient of cell cycle exit is the expression of the proliferation marker Ki67 in the basal turn, and its absence from apex and middle turns, where a zone of not-proliferating cells, demarking the PSD, was distinctly notable (Fig. 1b).

We then assessed the expression of the inner ear prosensory domain marker SOX2 (Fig. 1c, d, f, h). At W8, we detected SOX2 expression in the floor of the cochlear duct, marking the whole PSD; p27Kip1 was found in cells within this SOX2-positive domain (Fig. 1c). Over the course of the next 4 weeks, SOX2 expression became more restricted to the developing organ of Corti, where it was distinctly associated with the zone of non-proliferating cells marked with p27Kip1 (Fig. 1d).

We also assessed the expression of the PSD and supporting cell marker LGR5[17,18]. Immunostaining showed no detectable expression at W8 becoming weakly evident at W10 and at W12 (Fig. 1f). In situ hybridization[19] confirmed the presence of LGR5 mRNA in the PSD region, in all cochlear turns at W9 (Fig. 1g). In addition, as seen in mouse[20] some LGR5 mRNA could be detected in the region of the lesser epithelial ridge/spiral ligament. Low levels of sparsely positive cells were detected also in the spiral ganglion region (Supplementary Fig. 1).

Hair cell differentiation was examined by immunostaining for the hair cell marker MYO7A. Faint MYO7A expression was detectable around W11 in a subset of SOX2-positive cells that by location and morphology can be assumed to be nascent hair cells, visible exclusively in the basal turn (Fig. 1f, h). At W12, MYO7A, expression became robust in the base and middle turn but not in the apex, in cells that co-expressed SOX2. In humans, hair cell differentiation therefore appears, like in mice[7], to follow a basal-to-apical gradient (Fig. 1i). Hair cells in the developing human vestibular sensory epithelium, detected by MYO7A immunostaining, were already clearly manifest at W10. At week 12 they showed the expression of ESPIN positive hair bundles as well as nuclear expression of BRN3C (Supplementary Fig. 2).

With the goal of identifying surface markers that can be used to prospectively isolate PSD cells from the developing human cochlear duct, we assessed the expression of epithelial markers EPCAM/CD326[21,22], MCAM/CD146[22], as well as surface markers for neural stem cells Integrinα6/CD49f[21], Prominin/CD133[23,24], and SSEA1/CD15[25] (Supplementary Fig. 3a). EPCAM immunostaining was distinctly associated with the cochlear duct (Fig. 1j), and a well-defined EPCAM-positive population could be identified by flow cytometry (Supplementary Fig. 3a). CD49f antibodies labeled cells of the cochlear duct, as well as neuroblasts in the young spiral ganglion. No specific staining was detectable for CD133, MCAM/CD146, and CD15/SSEA1, neither on sections nor by flow cytometry. While weak immunoreactivity was observed on cyosections for LGR5, we did not identify a distinct population with flow cytometry that would allow for cell sorting as previously demonstrated for the murine organ of Corti[17,18]. E-Cadherin[26] staining appeared less specific than EPCAM in marking the cochlear duct (Supplementary Fig. 3b).

We also assessed the expression of NGFR, also known as CD271. Human CD271 expression was reported as restricted to the inner pillar region at week 14–18[27]. We found a dynamically changing pattern of CD271 expression that was prominently associated with developing spiral ganglion cells at W8, but this expression decreased in the ganglion at W10 and was near background at W12 (Fig. 2a). In the PSD, CD271 was detectable at W10 in a subset of supporting cells. Starting from W12, it became restricted and upregulated in cells that are distinctively located where pillar cells would be expected based on p27Kip1 expression and overall tissue morphology (Fig. 2c, d). Co-staining for the neurofilaments βIII tubulin (TUBB3) showed that CD271-labeling in the cochlear duct is not caused by neurites entering the PSD, but rather by distinct cells (Fig. 2c).

**Human spiral ganglion development**. Markers for the developing spiral ganglion neurons were selected based on the studies mainly conducted in mouse[28]. Expression of p27Kip1 was detected in the spiral ganglion at all-time points analyzed (Fig. 1a–c, j and Supplementary Fig. 4b), indicating that the nascent neurons presumably exited the cell cycle before W8. In mice, spiral ganglion neurons exit the cell cycle between embryonic day E10.5 and E12.5[2,29]. Also in agreement with mouse literature[30], human spiral ganglion neurons at these stages

lacked SOX2 expression, which was restricted to a few presumptive non-neuronal cells in the ganglion (Supplementary Fig. 4b). The expression of peripherin (PRPH) and TUBB3 was used to identify spiral ganglion neurons (Supplementary Fig. 4a, c, d). Our results confirm that human spiral ganglion neurons express both neurofilaments already from W8 onwards[12,13,27]. Islet1 (ISL1) and GATA3 were detected in the spiral ganglion neuron nuclei between W8 and W12, and were also expressed in the cochlear duct as previously reported during the murine development[31,32] (Supplementary Fig. 4c, d). NEUROD was only detectable in the youngest spiral ganglion neurons assessed at W8 and was absent in older stages. Doublecortin (DCX) expression

was detected at all-time points in the ganglion, while nestin (NES), was observed both in the ganglion, as well as in the cochlear duct and surrounding tissue (Supplementary Fig. 4c).

**Transcriptional profiling of human inner ear cells.** We further analyzed the gene expression profile of the cochlear duct, the spiral ganglion, and the utricle using a microfluidic qRT–PCR array for 190 genes known to be expressed during mouse otic development and maturation of the organ of Corti. Two time points (W8.3 and W11.1) were analyzed for cochlear duct and utricle. For the spiral ganglion, two additional time points were included in the study (W9 and W11.8). Hierarchical clustering

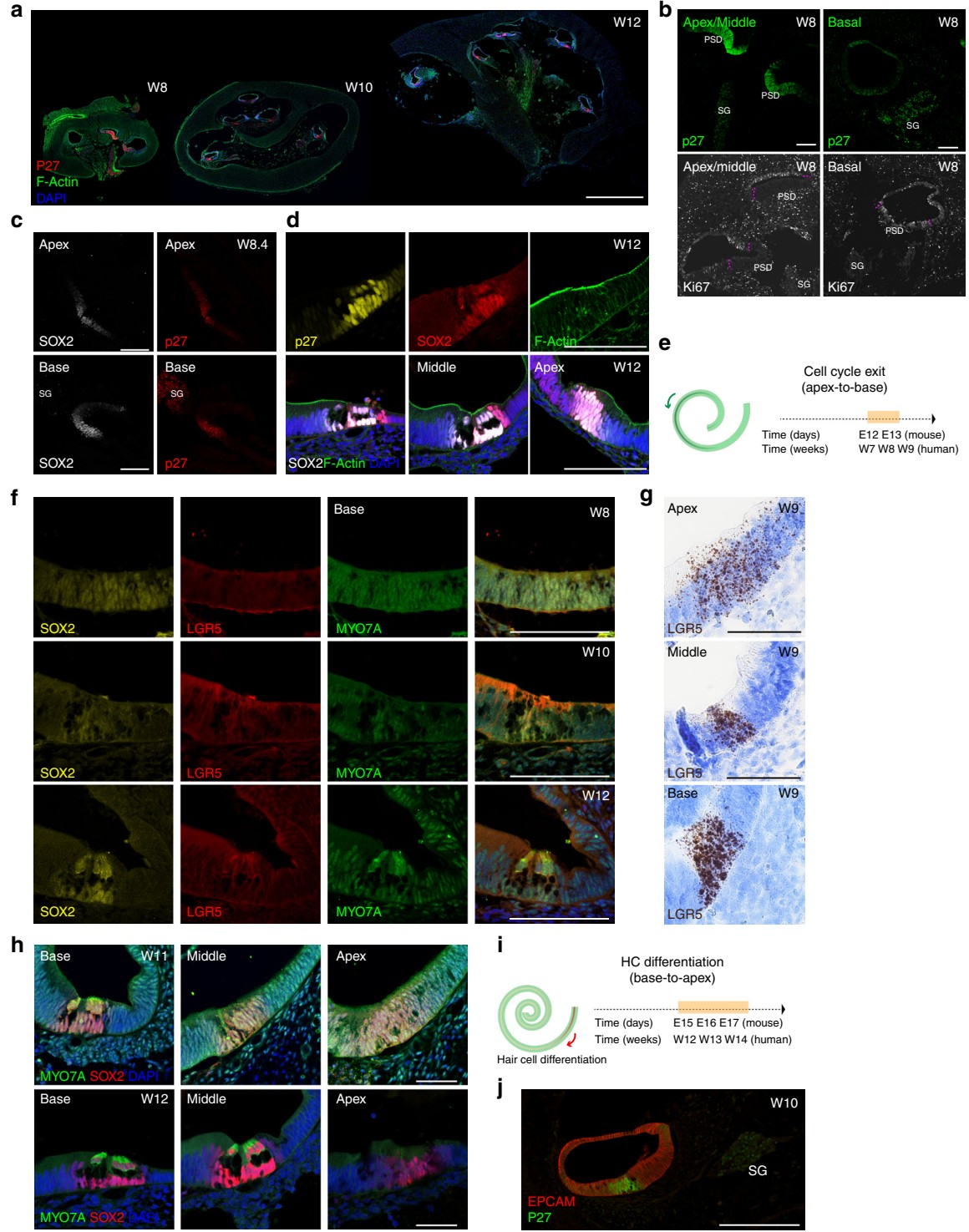

shows highest similarity between the two cochlear duct samples, the two utricle samples, and the four spiral ganglion specimen, independent of the developmental age. Cochlear ducts were more similar to utricles than to spiral ganglia (Fig. 3a). Markers for the early developing inner ear epithelia like *OC90, SOX9, FBXO2, EPCAM* and supporting cell markers were much stronger expressed in the cochlear duct and the utricle than in the spiral ganglia (Fig. 3b). Hair cell genes, like myosin 6 (*MYO6*), Usher1C (*USH1C*), *ATOH1, MYO15A, MYO7A, MYO3A,* Stereocilin (*STRC*), and Otoferlin (*OTOF*) were more strongly expressed in the utricle compared to the cochlear duct, which is compatible with our assessment that utricle hair cells differentiate before cochlear hair cells (Supplementary Fig. 2). When we compared the expression of sensory epithelium and hair cell genes in general between utricle and cochlear duct samples with spiral ganglion, we found that these genes, as expected, are more highly expressed in the developing sensory epithelia (Fig. 3c). Proneurosensory genes such as *GATA3* and *ISL1* were detected both in the cochlear duct, as well as in the spiral ganglia samples, in agreement with our histological assessment. Conversely, neuronal genes such as *TUBB3, NEUROD,* neurogenin 1, 2, and 3 (*NEUROG1–3*), were higher expressed in the spiral ganglion samples (Fig. 3d). Genes involved in inner ear and vestibular organ/semicircular canal development such as *HMX3, DLX3, DLX5* and *DLX6* were enriched in the utricle samples (Fig. 3e). Additional genes expressed during otic development such as *OTX1, OTX2,* and *EYA4* were enriched in utricle and cochlear duct and expressed at lower levels in the spiral ganglion; *SIX1, SIX4* and *EYA1* expression was less discriminate among the three different sample groups (Fig. 3f).

In general, the relative expression of each gene among the three different tissues followed the expression pattern one would expect, based on the accumulated knowledge gained in animal models[33–36]. We also did not encounter major differences with expectations for expression of Notch, Wnt, Sonic Hedgehog, and cell cycle genes, with the exception of *WNT2B* and *WNT3*, which were distinctively higher expressed in the human utricle samples, perhaps reflecting the more mature differentiation state of this organ compared to the rest (Supplementary data 1).

**Prospective isolation and organoid culture of cochlear duct derived cells.** Based on the specific expression of the epithelial cell surface marker EPCAM in the cochlear duct, we sorted EPCAM+ and EPCAM− cells from human cochlear samples between W9 and W12 of development (Fig. 4a). We consistently isolated 17,500–50,000 EPCAM+ cells per cochlea per sample, which represented $12.2 \pm 5.2\%$ ($\pm$s.d. $n = 9$) of the total cells.

To promote cell survival and recovery after sorting, we optimized a strategy for organoid formation exploiting cell re-aggregation in low-binding round-bottom 96-well plates in presence of Matrigel.

We first validated the protocol using Lgr5-GFP positive cells isolated from the sensory epithelium of early postnatal mice. In agreement with their previously reported hair cell progenitor characteristics[17,18] we were able to expand these cells and further differentiate them into hair cell-like cells (Supplementary Fig. 5). Like the murine cells, human EPCAM+ cells required addition of 2% Matrigel to the culture medium 2–3 days after sorting to enable the generation of epithelial organoids. In absence of Matrigel, the cells failed to grow when maintained in the same conditions. In contrast to the formation of organoids, we observed that the EPCAM− cells failed to form aggregates and developed mesenchymal/neuronal phenotypes instead, with some cells expressing nestin, βIII tubulin, and peripherin (Fig. 4b). When cultured for an extended time (56DIV) we occasionally encountered some neurons (Supplementary Fig. 6).

We analyzed organoids that formed from human cochlear duct EPCAM+ cell aggregates after 14 and 20 days in culture and found them well-structured with respect to epithelial organization and morphology; consistently showing expression of epithelial markers EPCAM, E-Cadherin (ECAD), and β-Catenin (CTNNB1), as well as CD49f. (Fig. 4c, green box). Many cells in the organoids were proliferating, without specific foci, as shown by scattered Ki67 staining; conversely we also found cells that had exited the cell cycle, indicated by p27Kip1 expression (Fig. 4c, red box). SOX9 and FBXO2, identified in the gene expression analysis as highly expressed genes and previously reported in human[12] and mouse development[35,37], were detected in all cells of each organoid tested (Fig. 4c, blue box). SOX2 expressing cells, used to identify putative PSD cells, were found as diffuse patches in some organoids (Fig. 4d, orange box). Lgr5 expression was not convincingly detectable (Fig. 4d). MYO7A staining did not reveal any cell with morphological characteristics of hair cell-like cells. Expression of the antigens listed here was assessed in primary as well as secondary organoids and no major changes in expression or localization of the respective proteins was observed with passaging (Fig. 4e, f). The organoids maintained proper epithelial organization as well as apical-basal polarity also in their second generation as shown by apical localization of the tight junction protein ZO-1 (Fig. 4e).

To promote cell proliferation in epithelial organoids, we tested the effect of adding the GSK3β inhibitor CHIR99021, which we had previously shown to enhance sphere forming capacity of murine Lgr5 cells[38], as well as promotion of growth of organoids derived from murine Lgr5+ sorted cells in our experiments (Supplementary Fig. 5d). Addition of CHIR99021 to the medium also lead to an increase in the size of the human organoids (Supplementary Fig. 5g). Organoids from EPCAM+ sorted cells have been successfully passaged in vitro by mild trypsinization and manual trituration, and cultured for up to 3 months

**Fig. 1** The human cochlear prosensory domain. **a** Three stages of human cochlear development (W8 (E1202), W10 (E1201), and W12 (E1210)). Shown are overview modiolar cyosections, immunolabeled with antibodies to p27Kip1. F-actin was labeled with phalloidin and cell nuclei were stained with DAPI. Scale bar = 1 mm. **b** Cochlea at W8 (E1202) of development, immunostained for p27Kip1 and Ki67. Right and left cochleae from the same fetus are shown. The prosensory domain (PSD) and the spiral ganglion (SG) are indicated. Pink dashed lines indicate the lack of KI67 positivity in the PSD in apical and middle turn. Scale bar = 100 μm. **c** Characterization of the W8.4 (E1251) PSD by immunostaining for SOX2 and p27Kip1. Scale bar = 100 μm. Apical and basal turns are shown as indicated. **d** Characterization of W12 (E1210 top and E1203 bottom) PSDs with immunostaining for SOX2 and p27Kip1. Upper panel shows the basal turn, lower panel shows the localization of the two markers in the base, middle, and apex. Scale bar = 50 μm. **e** Schematic comparison of the timing of cell cycle exit between mouse and human. **f** Immunostaining for SOX2, LGR5, and MYO7A of human fetal cochlear PSDs at W8 (E1202), W10 (E1201), and W12 (E1210) of development. Basal turns are shown for all-time points. Scale bar = 100 μm. **g** In situ hybridization (brown dots) with a LGR5-specific RNAscope probe at W9.2 of development (E1276) Basal, middle, and apical turns are shown. Sections are counterstained with haematoxylin (blue) to visualize tissue morphology. Scale bar = 50 μm. **h** Immunostaining for SOX2 and MYO7A of cochlear PSDs at W11 (E1195) and W12 (E1203). The basal, middle, and apical turns are shown. Scale bar = 50 μm. **i** Schematic comparison of the timing of hair cell differentiation between mouse and human. **j** Immunostaining of the entire cochlear duct for EPCAM (red) and p27Kip1 (green) at W10. Scale bar = 100 μm

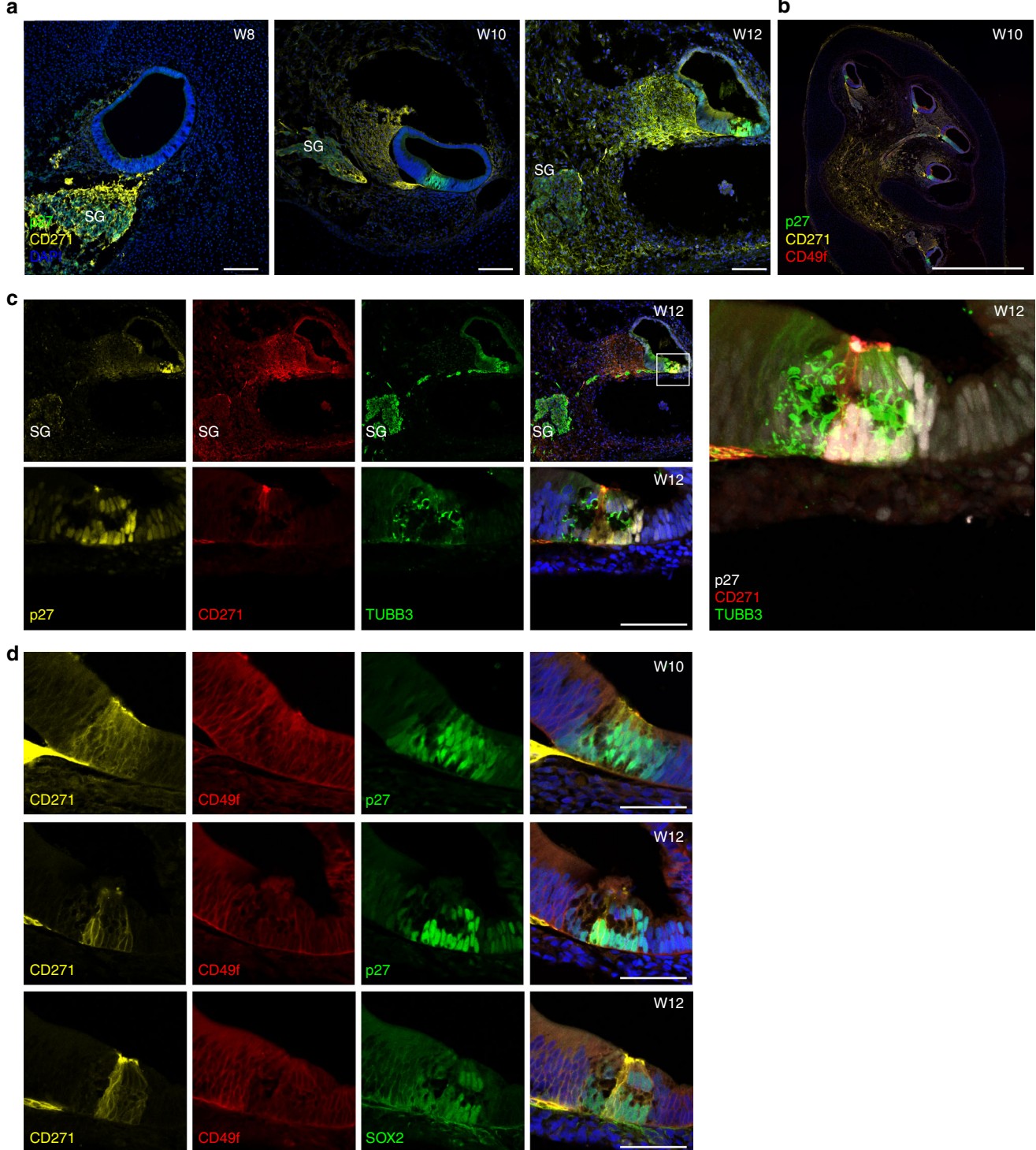

**Fig. 2** CD271 is expressed by epithelial cells within the prosensory domain. **a** Immunostaining for CD271 and p27Kip1 at three stages of development (W8 (E1202), W10 (E1201), and W12 (E1210)). Basal turns are shown. Scale bar = 100 μm. **b** Representative modiolar overview section immunolabeled with antibodies to p27Kip1, CD271, and CD49f (E1201, W10) scale bar 1 mm. **c** CD271, βIII Tubulin (TUBB3), and p27Kip1 at W12 of development. Overview pictures and higher magnifications are shown. Scale bars = 100 μm (top) and 50 μm (bottom). The enlarged right panel shows the merged confocal Z-stacks projected into a single image. **d** CD271, p27Kip1, SOX2, and CD49f immunolabelings at W10 (E1201) and W12 (E1210). Scale bars = 50 μm

(4 generations); however, the differentiation potential has not been assessed beyond the first generation in this study.

**Organoid differentiation to hair cells**. For assessing cell differentiation, we first grew EPCAM+ organoids for 2 weeks in presence/absence of CHIR99021, and then transferred the organoids into co-culture with the EPCAM− population (Fig. 5a). We utilized semi-permeable inserts that allow for exchange of conditioned medium, but no direct contact between the cells present in the two compartments. Simultaneously, we assessed the effect of the γ-secretase inhibitor LY411575, previously described to

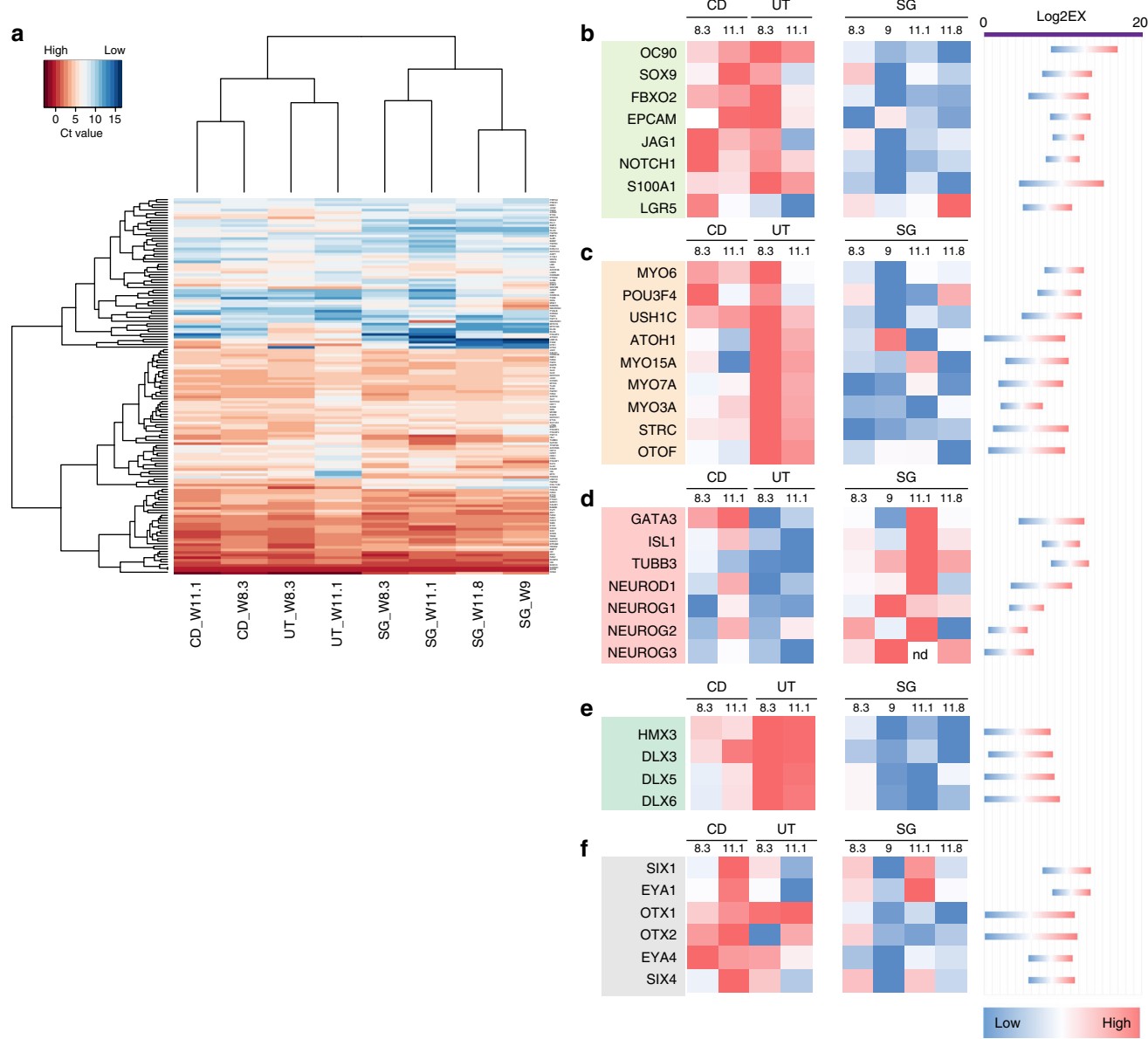

**Fig. 3** Gene expression analysis of the microdissected cochlear duct, utricle, and spiral ganglion. **a** Ct values obtained with multiplex qRT–PCR were used for hierarchical clustering. Strongly expressed genes with mean normalized Ct values below 10 are shown. Levels of expression are indicated by the color code. **b**–**f** Relative expression level of each gene among the three different tissues across multiple samples. The color code shown is a red-to-blue gradient binned into eight discrete values, with dark red indicating the highest expression level among the samples and blue the lowest. The range of expression values for each gene represented by the color range is shown on the right for each gene as Log2EX value (limit of detection) -the raw Ct value. **b** Epithelial and supporting cell markers; **c** hair cell markers; **d** neuronal genes; **e** vestibular organ development; **f** early otic development. Samples used for these experiments were E1204, E1208, E1235 and E1236

promote hair cell differentiation at the expense of supporting cells in the neonatal organ of Corti and in otic spheres derived from this tissue[39–42]. The cells were maintained in co-culture for ~2 weeks to reach in vitro the corresponding in vivo developmental age of 14–16 weeks. After this time period, we identified cell groups of MYO7A+/SOX2+/EPCAM+ cells in organoids derived from EPCAM+ sorted population, indicative of nascent hair cell-like cells (Fig. 5b).

We evaluated the efficiency of differentiation in organoids grown from five independent fetal samples, isolated from W11 fetuses (mean sample age: week 11.14 ± 0.45). The number of hair cell-like cells (positive for MYO7A and with a cytomorphology reminiscent of in vitro-generated hair cells[14,43] per organoid was on average 6.5 ± 6.6 (n = 11) in untreated samples, ranging from

zero to maximal 19 hair cell-like cells per organoid. Additionally, we observed F-actin-rich protrusions of hair cell-like cells in organoids indicative of putative hair bundles (Fig. 5c, d). Organoids treated with the GSK3β inhibitor CHIR99021 did not show a difference in the number of hair cell-like cells compared to untreated samples (Fig. 5e). LY411575 treatment increased the number of MYO7A-positive cells in some samples but not in others, an effect that did not reach statistical significance, and in combination with the GSK3β inhibitor, a lower number of hair cells was observed (Fig. 5e). As all sorted cells from each human sample were divided among the four experimental treatments and no significant changes were observed upon compound treatment, we estimated the total number of hair cells obtained for each fetal sample as the sum of

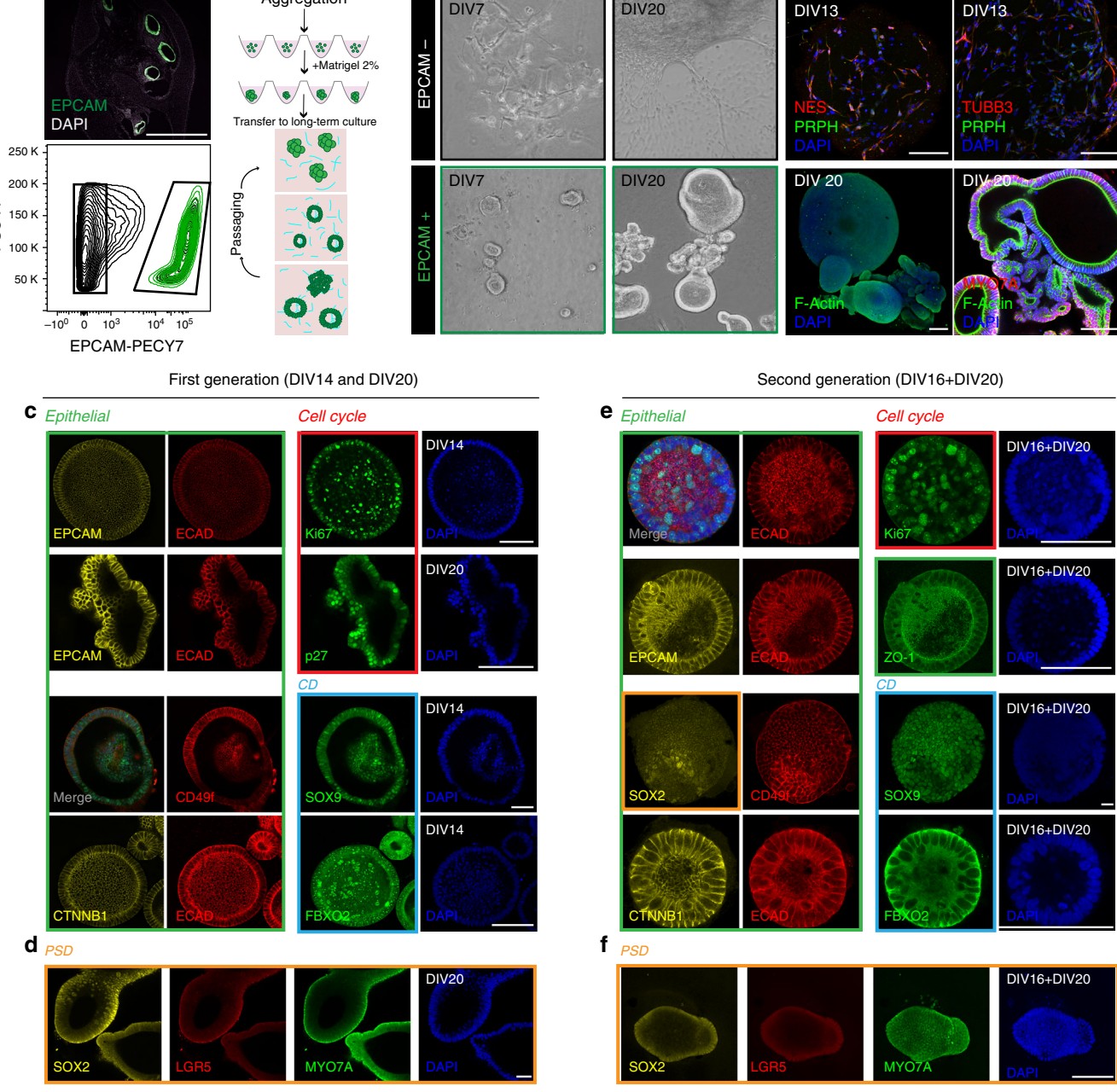

**Fig. 4** Organoid generation and characterization. **a** Human fetal cochlea at W10 (sample E1201) immunostained for EPCAM and FACS plot showing the gating of EPCAM-positive cochlear duct cells. Also shown is a schematic overview of the procedure for organoid generation. **b** Representative examples of EPCAM+ and EPCAM− derived cultures at day 7 (DIV7) and day 20 (DIV20) in vitro (sample E1220). EPCAM negative cells immunostained at day 13 for nestin (NES), peripherin (PRPH), and βIII Tubulin (TUBB3). EPCAM-positive organoids immunostained at day 20 in vitro for MYO7A. F-actin was labeled with phalloidin. Scale bar = 100 μm. **c–d** Immunostaining of 1st generation organoids (E1220 and E1253) at DIV14 and DIV20 for the proteins indicated. Colored boxes indicate marker classes: epithelial (green), cell cycle (red), cochlear duct (light blue), and PSD markers (orange) are shown. Scale bar = 100 μm. **e–f** Immunostaining of 2nd generation organoids (E1224) at DIV16 + DIV20 (days 1st generation + days 2nd generation) for the proteins indicated; marker classes are indicated by the colored boxes. Scale bar = 100 μm

hair cells of the four groups. The number varied from 20 cells to maximal 180 MYO7A+ cells (n = 45 organoids assessed).

**Cell sorting strategy to isolate human inner ear PSD cells.** Based on the expression pattern of CD271 in the PSD (Fig. 2), we assessed whether it would be feasible to use this marker, in combination with the surface antigen EPCAM to isolate PSD cells

from the developing human cochlear duct. Because co-expression of CD271 and EPCAM began during week nine of development, we focused on samples obtained between W9 and 12 for these experiments. Flow cytometry confirmed the presence of a double-positive population representing 2.3% ± 1.2 (n = 8 samples) of the total cochlear cell pool (Fig. 6a, b).

The use of two markers for cell sorting should in theory allow for simultaneous isolation of PSD cells, the remaining

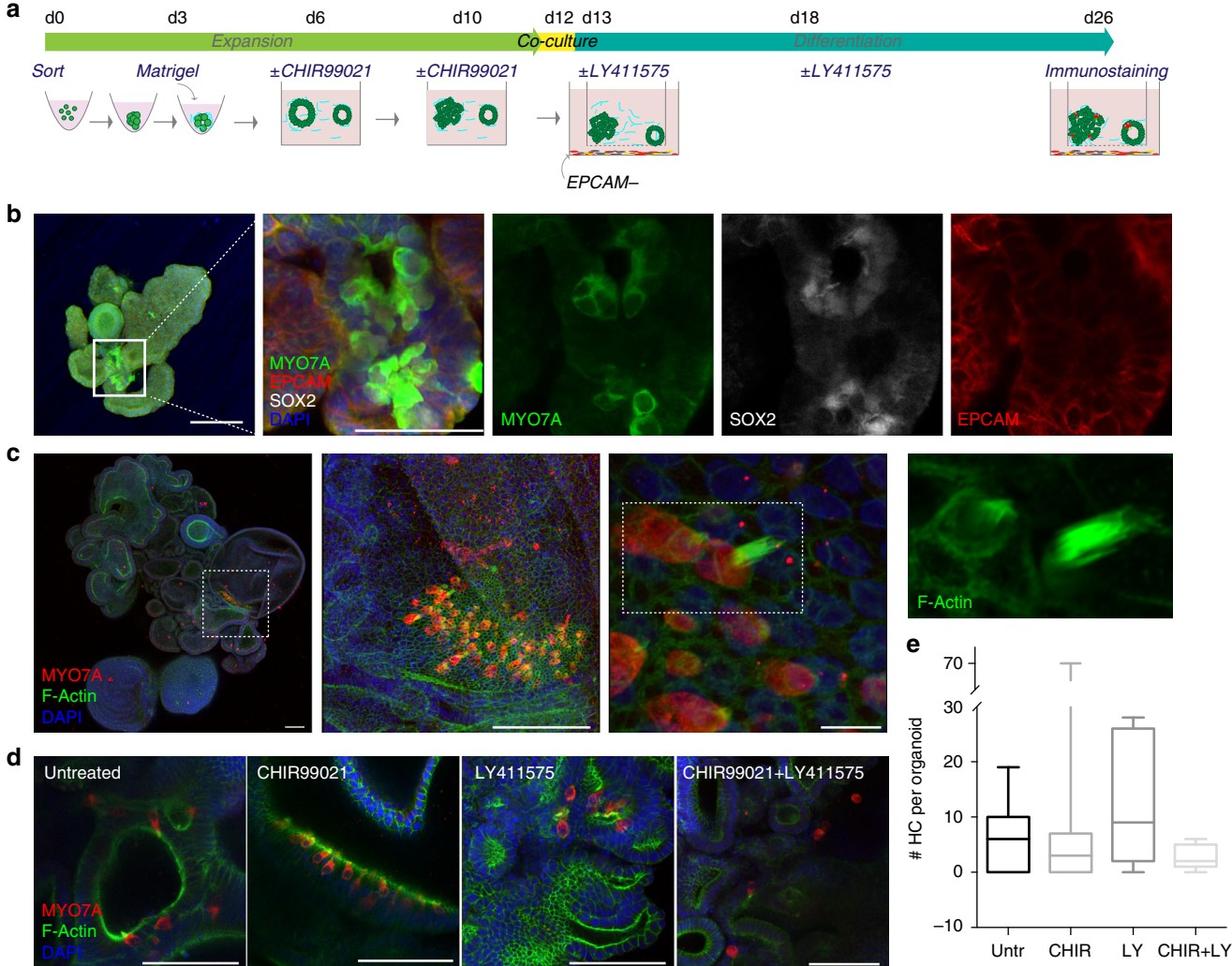

**Fig. 5** Organoids from EPCAM+ cells generate hair cell-like cells. **a** Schematic drawing of the experimental procedure. EPCAM+ organoids were expanded for 2 weeks with or without CHIR99021, then co-cultured with EPCAM− cells in transwell inserts. EPCAM− cells were derived in parallel from the same sample. The medium was supplemented twice with LY411575 or with vehicle during the 2-week differentiation period as indicated. **b** Organoid derived from sample E1229 and immunostained at the end of the differentiation protocol for MYO7A, SOX2, and EPCAM. Scale bar = 100 μm. The merged high magnification image is a confocal projection; the single channels are shown as individual confocal sections. Scale bar = 50 μm. **c** Organoid derived from sample E1246 (+CHIR99021) and immunostained at the end of the differentiation protocol for MYO7A. F-actin bundle was labeled with phalloidin. Consecutive magnifications are shown. Scale bars = 100 μm (left and middle), and 10 μm (right). **d** Images of MYO7A positive areas in organoids untreated/treated with CHIR99021 during the expansion phase and untreated/treated with LY411575 during the differentiation phase, as indicated. Scale bar = 100 μm. **e** Quantification of the number of hair cells-like cells (HC) per organoid at the end of the differentiation protocol for the different culture conditions as indicated. Three to four organoids per condition per sample were analyzed. Five human fetal samples of ≈week 11 were used for this experiment (E1245, E1246, E1229, E1248, E1253). Box plots showing sample distribution and median. Whiskers represent minimum and maximum values. Untreated: 6.45 ± 6.6 (n = 11); CHIR99021: 9.09 ± 19.84 (n = 11); LY411575: 10.46 ± 10.11(n = 13); CHIR99021 + LY411575: 2.53 ± 1.995 (n = 15). Values are mean ± s.d.

cochlear duct region, as well as the remaining mesenchyme/glial/neuronal population. Both EPCAM+ populations (EPCAM+/CD271− and EPCAM+/CD271+) formed as expected epithelial organoids (Fig. 6d, e and Supplementary Fig. 7b). The double-negative population (EPCAM−/CD271− cells) appeared as round cells incapable of substrate adhesion or cell–cell adhesion and did not survive beyond 5 days in culture. The EPCAM−/CD271+ population grew in Matrigel displaying a mesenchymal morphology (Supplementary Fig. 7c). We were able to expand this latter population and found that the cells expressed the mesenchymal marker vimentin and showed neither neuronal morphology nor

specific immunoreactivity for βIII tubulin (Supplementary Fig. 7d).

We induced differentiation to hair cell-like cells in the epithelial organoids again by co-culturing them in transwell inserts, this time with the mesenchymal EPCAM−/CD271+ cells. We consistently detected hair cell-like cells in organoids that formed from EPCAM+/CD271+ cells in 19 out of 19 organoids analyzed from five different donors (W9.7, W10.5, W11, W12, and W12.5) (Fig. 6 and Supplementary Fig. 8). Conversely, organoids derived from sorted EPCAM+/CD271− cells only occasionally gave rise to hair cell-like cells (n = 47 organoids from five different donors). Overall, 86.8+/−11.7% of the hair

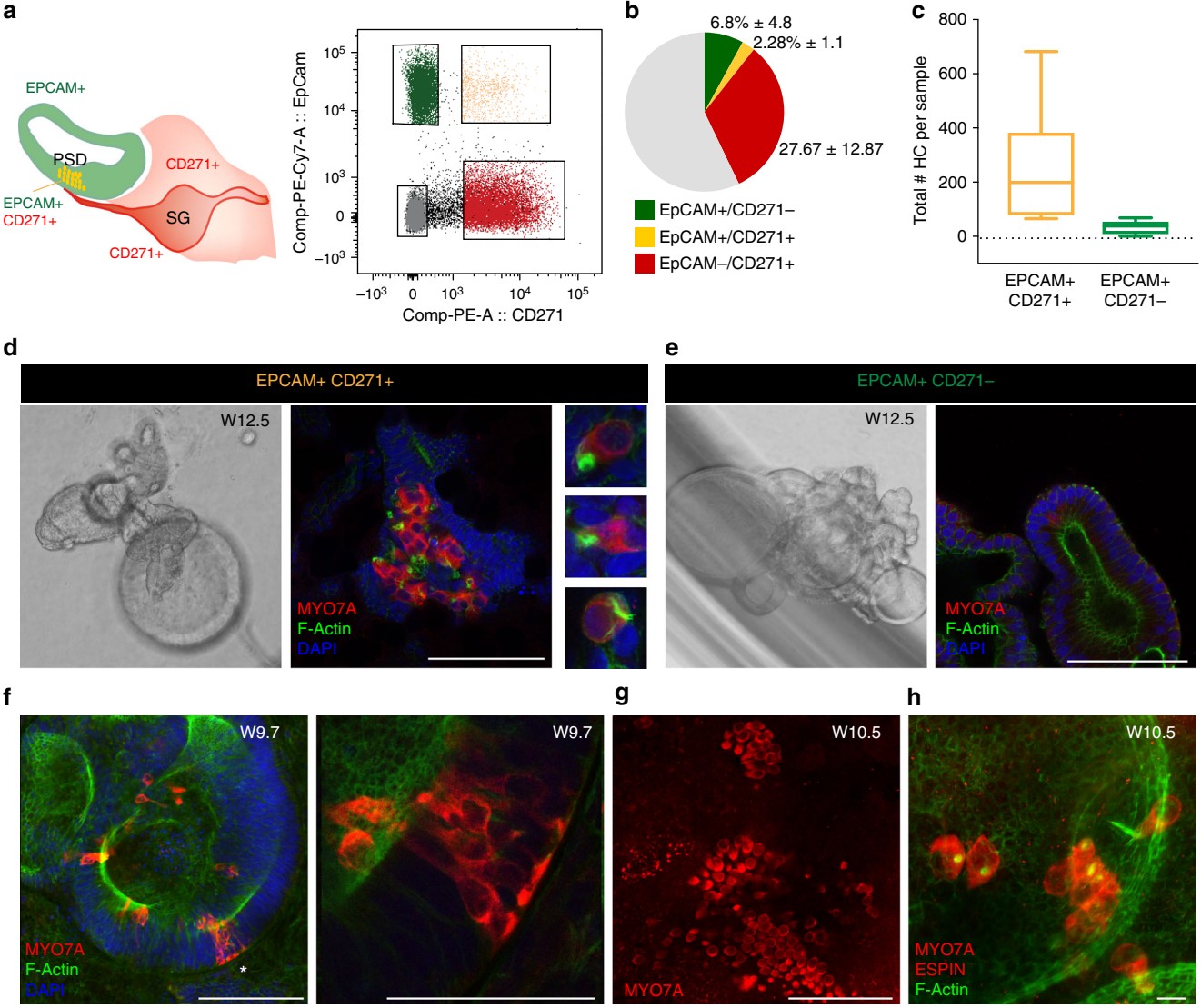

**Fig. 6** EPCAM+/CD271+ cells are hair cell progenitors. **a** Schematic illustration of EPCAM and CD271 expression in the cochlear duct. Double-positive cells in the PSD are shown in yellow. FACS plot showing the gated populations from a W12 sample (E1242). **b** Pie chart illustrating the distribution of the four FACS-sorted populations. n = 8 fetal tissues (week 9.4 to 12.3). Values are mean percent ± s.d. **c** Quantification of the total number of hair cells-like cell (HC) per sample obtained from the two sorted populations (N = 5 samples: W9.7 (E1238), W10.5 (E1270), W12.5 (E1242), W11 (E1254), and W12 (E1289)). EPCAM+/CD271+ cells: N = 19 organoids; EPCAM+/CD271− cells: n = 47 organoids.. Box plot indicates distribution, middle line: median, whiskers: minimum and maximum. **d** Representative examples of organoids derived from EPCAM+/CD271+ and **e** EPCAM+/CD271− populations (W12 sample, E1242). Immunostaining for MYO7A and co-staining with phalloidin and DAPI is shown. Selected hair cells-like cell morphologies from the 3D stack are shown in **d**. Scale bars = 100 μm. **f** Organoid derived from EPCAM+/CD271+ cells from a W9.7 sample (E1238) stained for MYO7A, F-actin and DAPI. Scale bar 100 μm. The area marked by the asterisk is enlarged in the right panel. **g** Organoid derived from a W10.5 sample (E1270) immunostained for MYO7A. Scale bar = 100 μm. The same organoid is shown at higher magnification in **h**. **h** The sample is re-stained for ESPIN (red) and F-actin. Scale bar 10 μm

cells-like cells was derived from the double-positive population (Supplementary Fig. 8a). These results support our hypothesis that CD271+ cells in the cochlear duct represent hair cell progenitors in the human fetal cochlea.

The number of human hair cell-like cells that formed in organoids from EPCAM+/CD271+ cells varied substantially, from a minimum of 38 to a maximum of 300 (Fig. 6c), but was significantly greater than what obtained from EPCAM+ cells only (as shown in Fig. 5e). Higher number of hair cell-like cells were observed in organoids that contained some mesenchymal cells as a consequence of a less clearly demarked population when isolated by FACS (Supplementary Fig. 8d).

Staining with F-actin revealed the presence of hair bundle protrusions in several of MYO7A positive cells in the EPCAM+/CD271+ organoids. (Figs. 6d, h, and 7). When tested for immunoreactivity with an ESPIN-specific antibody, we observed some of the more prominent bundles to be positive for this marker (Fig. 7a–c). Despite low in number, the hair cells-like cells identified in the organoids derived from the EPCAM+/CD271− population, also expressed ESPIN+ and F-actin+ bundles (Supplementary Fig. 8c).

As a surrogate readout for functionality, we assessed the uptake of the FM1–43 dye, known to permeate cells with active mechanotransduction channels[44–46]. To avoid aspecific dye

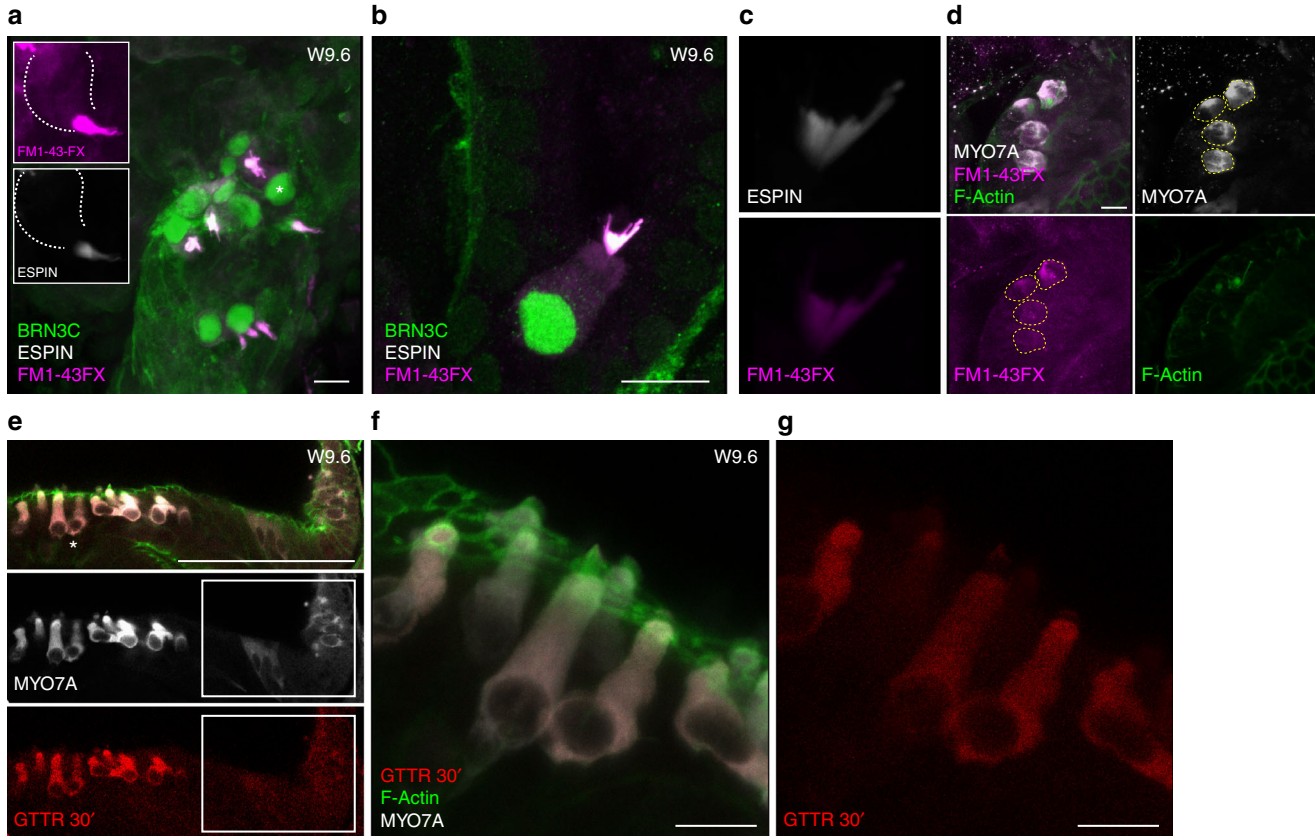

**Fig. 7** Functional characterization of the in vitro-derived hair cell-like cells. **a** Organoid derived from a W9.6 sample (E1286), stained for ESPIN and BRN3C after FM1–43 labeling. Scale bar 10 μm. The same cell marked by the asterisk is shown in the inserts. **b** Hair cell-like cell stained for ESPIN and BRN3C after FM1–43 labeling. Scale bar 10 μm. Gain values for FM1–43 are set in order to visualize intracellular dye loading. **c** Re-image with lower gain of ESPIN+ bundle and FM1–43. **d** Hair cell -like cells derived from a EPCAM+/CD271+ organoid from a W12 sample (E1289) stained for MYO7A and F-actin after FM1–43 loading. Cell volumes defined in the MYO7A channel (dashed lines) were used to assess fluorescence intensity in the FM1–43 channel as shown in Supplementary Fig. 9. Scale bar 10 μm. For all examples (**a**–**d**) FM1–43 uptake was performed with a 30 s exposure in presence of concanavalin A. **e** Hair cell-like cells in an organoid derived from an EPCAM+/CD271+ sample (W9.6, E1286) showing GTTR uptake. The sample was incubated for 30 min with GTTR, followed by fixation and immunostaining for MYO7A and staining with phalloidin to visualize bundles. 3D projection of a confocal stack is shown for the single channel and merged colors. Boxed area indicates developing MYO7A positive cells. Scale bar = 100 μm. **f** 3D reconstruction of the area marked in **e** by the asterisks. Merged image **f** and red channel **g** are shown. Scale bar 10 μm

uptake by endocytosis, organoids were pretreated for 10 min with a general blocker (concanavalin A)[46] and then co-incubated for 30 s with 5 μm FM1–43. This resulted in a strong accumulation of the dye in the hair bundle and cytoplasm of the ESPIN+/BRN3C + hair cell-like cells (Fig. 7a–c). The uptake could also be inhibited by Curare, previously shown to inhibit MET channels[46] (Supplementary Fig. 9a, b).

In addition, we assessed the capacity of the generated cells to uptake aminoglycosides, specifically a Texas-Red conjugated form of gentamycin sulfate (hereafter GTTR)[45]. Within the same organoid, we could identify more mature cells, showing hair cell-like morphology, F-actin rich bundles as well as displaying GTTR uptake (Fig. 7e–g and Supplementary Fig. 9c, d) along with neighboring developing, immature hair cells, already showing expression of MYO7A, but lacking functional bundles (Fig. 7e boxed area).

Despite the differences in cellular morphology and bundle morphology/maturity, we robustly and reproducibly obtained hair cell-like cells in organoids derived from EPCAM+/CD271+ sorted cells. The differentiation from cells isolated at early developmental stages (W10.5 and W9.7/9.6), thus several weeks prior to hair cell differentiation in vivo, demonstrates that these are generated in vitro from a pool of hair cell progenitors.

## Discussion

Loss of hair cells or spiral ganglion neurons accounts for the majority of cases of hearing loss and deafness. Reactivation of signaling pathways active during cochlear development could be exploited for induction of hair cell regeneration in the adult organ of Corti[47]. A substantial gap of knowledge, however, remains on human inner ear development, which ultimately will be the target of regenerative therapies.

We show that the human cochlear duct follows an apical-to-basal gradient of cell cycle exit while hair cell differentiation follows a basal-to-apical gradient[7,16]. We also show that neurons enter the cochlear prosensory domain before hair cell markers are detected. In addition, we describe the appearance of human vestibular hair cells prior to cochlear hair cells which is in agreement with previous reports[12,48–50]. Expression of prosensory domain markers (SOX2, LGR5, and CD271) and neuronal development markers (ISL1, GATA3, NEUROD, DCX, and NES) assessed in this study are in line with the available animal model literature. Our comprehensive gene expression profiling of the developing human inner ear comprises a gene panel that covers both early and late stages of otic development. Our results agree with previously generated murine datasets[34,36,51,52] and support the findings of our immunohistological assessments.

These datasets are of particular interest to provide a blueprint of inner ear development and guide efforts into deriving in vitro inner ear sensory cell types from pluripotent progenitors[14,53,54]. Additional studies focusing on the direct comparison between the transcriptional profiles of the sorted somatic human fetal progenitors or in vitro-derived hair cells from these sources, and the human IPSC/ESC derived hair cell/progenitors will be key to reveal to which extent pluripotent cells recapitulate human development in vitro[55].

We report moreover on a novel approach to prospectively isolate, expand in vitro and finally differentiate into hair cell-like cells the sorted human somatic progenitors. The use of three-dimensional culture conditions for cells of epithelial origin has been demonstrated to be particularly suitable to preserve tissue organization, cellular interactions, and tissue development in different organs[56]. Recently this has been applied successfully to murine postnatal cochlear progenitors[57] as well as for the differentiation of hair cells from pluripotent stem cells[14,54,58]. The hair cell-like cells generated in this study displayed markers, morphology, bundle expression, and properties of hair cells that were not previously achieved using 2D culture on stiff substrates from similar cell sources[59,60].

Despite their postmitotic state in vivo at the time of isolation, cochlear duct resident cells were induced to proliferate in vitro, as shown by Ki67 staining, by the pronounced expansion of the organoids, and by the proliferative response to GSK3β inhibition. The latter possibly dependent on canonical Wnt signaling, as expected based on the expression of the Wnt co-receptor and target LGR5 in the human cochlear duct[17,18,57]. Despite LGR5 mRNA expression, we could not rely on this antigen for FACS sorting of PSD resident cells. Instead, we found that the surface markers CD271 and EPCAM are well suited for isolation of human inner ear PSD cells. CD271 has been previously used for isolation of supporting cells with hair cell progenitors characteristics in mice[61].

The pool of sorted cells likely comprises a heterogeneous cell population, which varies with the developmental stage, as suggested by the changes in expression of CD271 in the PSD. We speculate that earlier developmental stages may yield to the isolation of less committed cells. In contrast, isolation of double-positive cells from late stages, will result in the isolation of a restricted population of supporting cells, which may have a more limited potential.

Recently described culture conditions using small molecule Wnt activation and concomitant chromatin modulation by HDAC inhibition[57] could represent promising means to alter the potential pre-committed fate of the isolated human fetal cochlear duct cells, induce their expansion and further differentiation. Alternatively, approaches to immortalize or partially reprogram[62–64] the somatic progenitors, may enable the generation of human progenitor lines, amenable to expansion, which could eventually provide powerful experimental systems for in vitro drug validation and screening.

An additional key requisite for the latter will be the reliable generation of functional hair cells. Currently a subset of the differentiated hair cell-like cells displayed F-actin, as well as ESPIN rich protrusions on their apical surfaces when assessed after 4–5 weeks in vitro. We could also observe uptake of the styryl dye FM1–43 in culture. The experimental conditions used, (short incubation time and co-treatment with the endocytosis blocker concanavalin A) allowed to detect also robust signals at the level of the hair bundle, suggesting the compound may be entering through the stereocilia as previously demonstrated[44–46]. Similarly, gentamycin Texas-Red was uptaken by cells displaying F-actin bundles, but not by more immature cells. The three-dimensional organization of the cells and the large size of the organoids did not allow to identify hair cells nor their bundles by light microscopy and probe electrophysiologically the mechanosensitive properties of the in vitro-generated hair cells. These studies will rely on the future generation of fluorescent reporter lines, by knock-in strategies[14], which will also serve as a tool to optimize culture conditions.

Implementation of the culture system with bioreactors, or non-static flow condition may implement the health of the culture, as shown in other systems[65,66], as well as it may impact on hair bundle development and functional maturation and increase the yield of functional hair cells.

In conclusion, our study provides a detailed analysis of the developing human inner ear, specifically of the cochlea and contributes a valuable dataset to comparative neurobiology. Moreover, it establishes the fundamental principles for the purification of human cochlea sensory hair cell precursors, as well as new methodology for their in vitro culture, expansion and differentiation to hair cell-like cells.

While the yield and scalability to date lags behind the potential of (induced) pluripotent cell derived sensory cells, this study lays the foundation for a future systematic comparison of the two strategies. Native human cochlear progenitors and hair cells derived from these, offer a benchmark to implement differentiation protocols from more abundant stem cell sources for efficient generation of cochlear hair cells.

## Methods

**Tissue isolation, staging, and dissection**. The inner ear was isolated from aborted human fetuses ranging from W8 to W12 post conception. Signed informed consent of the donors for procurement of the aborted fetuses and for use of tissues in research was obtained (after the donors' decision to terminate pregnancy, prior to the procedure). Donors were provided with the necessary information about the research project by medical staff with no vested interest in the research protocol. Only afterwards the research team was informed of the donation. Procurement and procedures were performed with full approval by the Ethics Committee of the Medical Faculty of the University of Bern and the Ethics Committee of the State Bern, Switzerland (Gesundheits-und Fürsorgedirektion des Kantons Bern, Kantonale Ethikkommission für die Forschung (Project ID: 2016–0033/KEK-Nr. 181/07). All experiments including the procurement and processing of human fetal tissue or organs were performed in accordance with guidelines enunciated in the current version of the Declaration of Helsinki (DoH) and the Essentials of Good Epidemiological Practice issued by Public Health Schweiz (EGEP). Tissues were collected the same day, as early as possible after the procedure, otherwise discarded. Samples were then anonymised (EXXXX). Damaged cochleas, with blood inclusions or visibly broken were not included in the analysis. Supplementary Table 1 lists all specimens used in this study. Fetuses were carefully staged according to generally accepted guidelines[67]. The foot length, distal, and proximal arm length and distal and proximal leg length were used to calculate the days post conception (p.c.). The number of days p.c. was then divided by seven to calculate the week of development. When this was not possible, the postmenstrual date was used for the calculation. Tissue dissection was performed in ice-cold Hanks' balanced salt solution (HBSS). The utricle was isolated from the vestibule. After removal of the cartilageneous/bony capsule of the cochlea, the cochlear duct was removed using fine tweezers and the remaining modiolar tissue, containing the spiral ganglion, was collected. For sorting experiments, the spiral ganglion, modiolus, and cochlear duct were collected without further microdissection. Alternatively, the whole-inner ear was fixed with paraformaldehyde (PFA) for cryosectioning.

**Cryosectioning**. Samples were fixed with 4% PFA in phosphate buffer saline (PBS) for 24 h at 4 °C, or for 2 h at room temperature. After washing with PBS, the samples were decalcified in Osteosoft (Millipore) for 2 weeks. Specimens were then washed 3X with PBS, incubated for 2 h in a solution of 15% sucrose, and subsequently overnight in 30% sucrose. Finally, the samples were placed for 30 min in optimal cutting temperature (OCT) compound (Sakura Finetek, USA), and snap frozen in a dry-ice/ethanol bath. Sections of 16 µm were cut with a cryostat (Leica CM3050 S).

**Immunostaining and imaging**. For immunostaining, the samples were permeabilized by treatment for 5 min with 0.1% Triton X-100 (Sigma) in PBS and blocked with 2% bovine serum albumin (BSA, Sigma) in PBS with 0.01% Triton X-100 for 2 h. Primary antibodies were incubated overnight at 4 °C and samples were then washed 3X with PBS, followed by 2 h incubation with secondary Alexa-fluor labeled antibodies (Invitrogen/Thermofisher), diluted 1:500 in PBS with 2% BSA and 0.01% Triton X-100 at room temperature.

For whole-mount staining and imaging of the utricle as well as for organoids, permeabilization was conducted with 3% Triton X-100, blocked for 2 h with 2% BSA and 0.01% Triton X-100 in PBS, and subsequently incubated with primary antibodies for 72 h at 4 °C. Samples were then washed with PBS, incubated with secondary antibodies for 72 h, washed and analyzed by confocal microscopy. Samples were first imaged as unmounted/floating samples to obtain low magnification (×10 and ×20) overview images. Subsequently, the specimens were mounted on glass slides with Fluorshield-with DAPI (Sigma) and re-imaged with ×63 or ×40 oil objectives. Images were acquired with a LSM710 confocal microscope (Zeiss). Deconvolution was performed using Huygens Remote Manager.

A list of all antibodies used in the study and the dilutions used is provided in Supplementary table 2.

**In situ hybridization (ISH)**. ISH for *LGR5* was performed on 5 μm sections of paraffin embedded fetal cochleas or intestine, using the RNAscope® LS 2.5 Probe Hs-*LGR5* (NM_003667.2, region 560–1589 cat#311028) using BOND RX (Leica Biosystems). Cochleae were fixed in 4% PFA after isolation for 24 h at 4 °C, then decalcified in osteosoft for 48 h and subsequently processed for paraffin embedding. The probe was purchased from ACD (Advanced Cell Diagnostics, Hayward, CA) and used according to the manufacturer's instructions. Briefly, samples were baked at 60 °C for 30 min followed by deparaffinization and pretreatment with Tris buffer at 95 °C for 15 min and protease for 5 min at 37 °C. Slides were incubated with the probe for 2 h at 40 °C, followed by successive incubations with Amp1 to Amp6 reagents. Staining was visualized with 3,3- diaminobenzine (DAB) for 10 min, then counterstained with haematoxylin. PPIB was used as positive control probe (ACD, NM_000942.4, region 139–989, catalog number 313908). The experiments were performed with the help of the Translational Research Unit (TRU) of the Institute of Pathology, University of Bern.

**Tissue isolation for gene expression analysis**. Tissue dissection was performed in ice-cold HBSS. Sterile forceps were cleaned with RNAseZAP (Ambion). The utricle was isolated from the vestibule. The cochlear duct was removed using fine tweezers and the remaining modiolar tissue, containing the spiral ganglion, was collected. Two different developmental stages were collected for utricle and cochlear duct (W8.3, W11.1). For the spiral ganglion, two additional samples (W9 and W11.8) were analyzed. Each sample was minced using Dumont #5 forceps and incubated in 100 μl extraction buffer from Arcturus PicoPure RNA isolation kit (Thermofisher). RNA isolation was performed in accordance with the kit manufacturer's instructions. Quantity of 200–1000 ng total RNA was used for cDNA synthesis using High Capacity cDNA Reverse Transcription Kit (Applied Biosystems). cDNA was diluted fivefold in water and used for target gene-specific pre-amplification. Volume of 1.5 μl of cDNA per sample was pre-amplified for 14 cycles with 500 nM DELTAgene pooled primer mix using 2X Taqman PreAmp Master Mix (Invitrogen), followed by Exonuclease 1 treatment (NEB). Fivefold diluted pre-amplified cDNA was used for loading the 96.96 Dynamic Array chip on the Fluidigm Biomark HD. A full list of primers used in the assay is provided in Supplementary data 2.

**Quantitative RT–PCR (qPCR) primer validation**. To validate DELTAgene primers, a two-fold dilution series spanning a gradient of 15 concentrations was performed on bulk RNA from human utricle. The mean Ct value for the most dilute sample in which positive amplification plots were detected in all six replicates with a standard deviation <1 was set as the limit of detection (LoD) for each primer pair. The overall LoD for the panel of assays was determined by taking the median of the LoDs for each assay, equal to a Ct of 21.

**Data analysis qPCR**. Samples were run in technical triplicate and the average Ct was used for the analysis. Ct values obtained by the qPCR were normalized to the housekeeping genes β-Actin and GAPDH. The data are presented as Ct values. Clustering analysis was performed with R-Studio for all genes, and for genes that had a mean Ct value below 10 (highly expressed genes). Alternatively, the Ct values for a panel of selected targets are color coded and displayed for each gene. The complete set of data is provided in Supplementary data 1.

**Single cell preparation for FACS**. The two cochleae, the spiral gangliae, and the modioliae were placed in drops of PBS, immediately after collecting, and carefully minced using forceps. Trypsin-EDTA was added to a dilution of 0.25% and the samples were incubated for 15 min at 37 °C. Trypsin activity was blocked by adding equal volumes of trypsin inhibitor mix, consisting of trypsin inhibitor from soybean (1 mg/ml) and DNAse1 (1 mg/ml), both from Sigma-Aldrich. Cells were mechanically triturated by pipetting. The cells were then washed with 5 ml ice-cold PBS, and centrifuged for 5 min at 190×g. The cell pellet was resuspended in 500 μl ice-cold sterile PBS containing 2% BSA, and the cell suspension was strained through a 40 μm filter (Falcon). Ninety percent of the cell suspension was used for sorting CD marker labeled cells, 10% was used for single staining, and negative controls. For CD marker labeling, the cell suspension was incubated for 30–45 min on a shaker in the dark, on ice, with 5 μl anti EPCAM-PE-Cy7 (Biolegends) antibody alone or in combination with 5 μl anti CD271-PE conjugated antibody

(BD bioscience). For initial testing of antibodies, 100 μl cell suspension was incubated with 1 μl fluorescently conjugated antibody on ice. Cells were then washed twice with 2% BSA in PBS, and sorted using a FACS ARIA equipped with a 100 μm nozzle. Cells were gated based on FSC-A and SSC-A to exclude debris, and subsequently FSC-A/FSC-H to exclude aggregates (representative examples are given in Supplementary Figures 3 and 8). Automatic channel compensation with single staining was used.

**Human inner ear organoid formation**. Two thousand FACS-sorted cells were plated per well into round bottom U-shaped low-adherent 96-well plates (Costar) in DMEM-F12 supplemented with B27, N2 (all from Life Technologies), 20 ng/ml EGF, 10 ng/ml bFGF, 50 ng/ml IGF, and 50 ng/ml heparan sulfate (Sigma-Aldrich). Two–three days after sort, Matrigel (GF-depleted, Corning) was added at a final concentration of 2%. Cells were incubated for 1 week at 37 °C and subsequently transferred, to low-adhesion 24-well plates (Costar) in medium with 2% Matrigel. The GSK3β inhibitor CHIR99021 (3 μM) was added to the culture at two time points (d5/6 and d10/11) together with supplementation of fresh medium.

**Human inner ear organoid differentiation**. Organoids expanded for 2 weeks were plated on transwell-permeable support membranes (Corning) in 12-well plates. 100 μl of 50% Matrigel was used to pre-coat each insert for 30 min before plating the organoids. This coating facilitated the immobilization of the organoids at the bottom of the insert. EPCAM-sorted cells were cultured in the lower compartments at 70%–100% confluency directly on plastic. Supplemented DMEM-F12 medium was further supplemented when indicated, with 1 μM LY411575 and left unchanged for 7 days. The medium was then refreshed by replacing 50% of the medium with fresh one (DMEM-F12 with B27 and N2, lacking growth factors) and re-adding LY411575 (1 μM) every 3–4 days. Organoids were fixed and immunostained for characterization as described above.

**FM1–43 and GTTR loading experiments**. Organoids were incubated for 30 min with Hoechst at 10 μg/ml in medium at 37 °C. FM®1–43FX (Invitrogen/Molecular probes), freshly prepared from a lyophilized stock, was added for 30 s at 37 °C at the concentration of 5 μM. After 30 s, the medium was quickly removed, and samples were washed with PBS and then fixed with 4% PFA in phosphate buffer saline (PBS) for 15 min at RT, washed twice and imaged by confocal microscopy.

Gentamicyn-Texas Red, GTTR (kind gift of Anthony Ricci, Stanford University) was dissolved from a dessicated stock at 30 mg/ml and used in vitro at the concentration of 0.3 mg/ml. Organoids were incubated for 30 min at 37 °C. Samples were washed with PBS and then fixed with 4% PFA in PBS for 15 min at RT, washed twice and imaged by confocal microscopy.

Subsequently, samples were incubated in 2% BSA in PBS containing 0.01% Triton X-100 for 1 h at RT and then stained overnight with primary antibodies at 4 °C. Secondary antibodies were added for 2 h at room temperature.

When indicated, Concanavalin A (Con-A, Sigma) or Tubocurarine chloride pentahydrate (Curare, Sigma) were added 10 min prior to the addition of FM1–43 at the concentration of 2.5 μM and 1 mM, respectively, as previously reported[46] to inhibit endocytosis (Con-A) and as MET channel blocker (Curare).

For the quantification of FM1–43 loading, cells were immunostained with MYO7A (secondary antibody alexa 647) and labeled with Phalloidin-ATTO-488. 3D stack were acquired with laser scanning confocal using sequential mode and filters for the PMTs were selected in order to avoid channel crosstalk. 3D images were then reconstructed and MYO7A positive cells were identified. Fluorescence intensity (mean gray value) for the FM1–43 channel was quantified in the volume identified by MYO7A staining using FIJI (https://imagej.net/Fiji). For each cell, background fluorescence was quantified in an immediately adjacent area and background was subtracted for each cell.

**Hair cell quantification in organoids**. Organoids were confocally imaged with ×10 and ×20 objective using optimal Z-stack spacing for each objective. Z and 3D projections of the images were generated with FIJI and hair cells were counted manually using the "cell counter" plugin. Montages of each plane were assessed in addition to double-check the software-assisted counting. Hair cells-like cells that were quantified were qualitatively defined by the expression of Myosin7a in the cytoplasm and nuclear exclusion of the staining.

**Organoid formation from murine Lgr5-GFP cells**. All mouse experiments were approved by the Animal Research Ethics Committee of the Canton Berne, Switzerland, (permission number BE42–15), and were carried out in accordance with the approved guidelines.

The sensory epithelium was isolated from mice containing an EGFP-IRES-CreERT2 knock-in allele at the Lgr5 locus, referred to as Lgr5-GFP (Jackson Labs Stock 008875). Three- to five-day-old neonates were used. Specimens from 7 to 10 animals were pooled for sorting experiments. Sensory epithelia were microdissected and triturated using fine forceps, placed in Trypsin-EDTA at a dilution of 0.25% and incubated for 15 min at 37 °C. Trypsin activity was blocked by adding equal volumes of trypsin inhibitor mix (Trypsin inhibitor from soybean (1 mg/ml) and DNAse1 (1 mg/ml)). Cells were then mechanically triturated by pipetting, washed with 5 ml ice-cold PBS, and centrifuged for 5 min at 190×g. Cells were then strained

through a 40 μm filter (Falcon) and sorted based on GFP expression. Positive and negative cells were plated in round bottom U-shaped low adherent 96-well plates (Costar) at 2000 cells per well in supplemented DMEM-F12 medium as described above. Three days later, Matrigel was added to the wells at the final concentration of 2%. Cells were incubated for 1 week at 37 °C and half of the medium was replenished once during this period. The GSK3β inhibitor CHIR99021 was supplemented on day 1 (10 μM). From day 9 to day 14, half of the medium was replaced with basic medium (DMEM-F12 with B27 and N2, without growth factors) every 3 days. Hair cell differentiation was assessed at day 15–20 for three independent experiments.

**Statistical analysis and reproducibility of the findings**. The samples used for each experiment are detailed in the figure legends and supplementary table 1. For the histological characterization, three samples of comparable age were used for each developmental week ($n = 2$ for W12). For ISH, two samples of similar age were used. Organoid differentiation from EPCAM+ cells was repeated for five human samples at W11 of development, using the same protocol. Statistical comparison between the treatment groups was performed using the non-parametric Mann–Whitney test (two-tailed), or unpaired $t$-test (two-tailed) for normally distributed samples, for each group compared to the untreated condition. Organoid differentiation from EPCAM+/CD271+ or EPCAM+/CD271− cells was performed with cells individually sorted from five donors at four different developmental stages. The number of organoid analyzed per condition is indicated in the figure legends.

## Data availability

All data generated or analysed during this study are included in this published article (and it supplementary information files).

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

## Acknowledgements

We thank the department of Obstetrics, Frauenklinik Bern, in particular the study nurse Ms. J. Wanner, for help with the tissue collection. Thanks to Dr. Di Santo for help with sample collection and staging. We're grateful to the flow cytometry core facility of the Department of Biomedical Research (DBMR) and the Microscopy Center at the University of Bern. Special thanks goes to Stefan Mueller, Thomas Schaffer and Bernadette Nyfeler for their flexibility and assistance with the experiments. We thank Irene Keller from the Bioinformatic unit of the DBMR for help with gene expression data analysis and hierarchical clustering and the Translational Research Unit of the pathology institute, in particular Dr. Galvan, as well as the group of Volker Enzmann, for help with ISH. We thank Dr. Mary O'Sullivan and prof. Anthony Ricci, Stanford University, for providing Gentamycin-Texas Red and prof. James Bartles, Northwestern University, for providing the espin antibody. We are grateful to all Otostem (www.otostem.org) partners for fruitful scientific discussions and suggestions. This project was sponsored by the EU-FP7 under the Health topic, grant number 603029.

## Author contributions

M.R. and P.S. conceived the study. M.R. designed the experiments. M.R., M.P., H.R.W. performed experiments, including tissue collection, histological characterization, confocal imaging, and cell culture. M.E. and S.H. performed experiments relative to gene expression profiling. M.R. analyzed data and wrote the paper. P.S. and S.H. contributed to data interpretation and manuscript writing. All authors had access to all the data, and approved the final version of the manuscript.

## Additional information

**Competing interests:** The authors declare no competing interests.

