## [Peer Review File · Nature Communications]

Reviewers' comments:

Reviewer #1 (Remarks to the Author):

Review of Roccio et al. Nature Communications

For this submission, Roccio and colleagues harvested inner ear tissue from aborted human fetuses between six and 12 weeks post-conception. The tissue was characterized using markers for the prosensory domain (Sox2), stem cell markers (LGR5), markers for cell cycle exit (p27Kip1) and markers that indicate emergence of hair cells (Myo7a). Use of these immuno markers has established the developmental sequence in the mouse cochlea, but how and whether these markers could be used to track human cochlear development was unknown. As such, the investigators established a critical period for human organ of Corti specification between week 8 and 12, which seems to parallel mouse inner development between, embryonic day 12 and 13.

The authors went on to use an qRT-PCR-based approach to characterize expression of 190 genes of particular interest in the inner ear and tracked expression at two developmental stages from three distinct regions. This analysis significantly extends the usefulness and impact of this work.

The authors also screened a series of cell surface markers in hopes of finding some useful for cell sorting. They identified EPCAM, which labelled cells of the developing cochlear duct. Sorted tissue yielded as many as 50,000 cells which were confirmed to be Lgr5-positive. The proliferative capacity of the cells was validated and there was confirmation using assays for additional markers. As expected, EPCAM- cells did not meet these criteria.

CD271 was used as a second marker to sort hair cell progenitors. Using double selection for both EPCAM and CD271 increased yield. In the second half of the manuscript, the authors used the EPCAM+ and /or CD271+ cells to generate inner ear organoids using what are now standard techniques. Hair cells numbers were low, but convincing.

The manuscript is well written and presented in logical easy to follow fashion. Figures are clearly labelled and the images and data are of excellent quality. In all these respects, the manuscript is consistent with the high standards of the Heller / Senn duo and importantly, this characteristic seems to be true of corresponding author, Marta Roccio, as well.

While the work is not hypothesis driven, it is a thorough first-of-its-kind of molecular and genetic characterization of human inner ear development, as well as, a demonstration that cells harvested from the fetal inner prosensory domain, can be used to generate human hair cells. The work will be an important point of reference for future studies aimed at reproducing human inner ear development to promote restoration of inner ear structure and function. As such, the manuscript will be of broad interest to inner ear biologists, as well as others interested in human development and regenerative medicine.

Below are listed a few additional points to consider that may further boost the impact of this

work.

- 1) Despite some optimization, the numbers of hair cells produced was low relative to other mouse and human organoid protocols. For this approach to be broadly useful, some further commentary on potential methods to boost yield would be helpful.
- 2) To confirm cell identity, hair cell markers are useful, but of course, the key identifying characteristic of a hair cell is the hair bundle, which is only shown for one cell, in one image. Even with the low yield, might there be other hair bundle images that could be presented to boost confidence in the approach?
- 3) The one hair bundle shown appears to be vestibular in nature, but was presumably derived from cells harvested from the cochlear duct. This appears to be a recurrent theme in organoid and other protocols used to generate hair cells. Could the authors comment on why a vestibular hair cell phenotype seems to be the predominate hair cell fate, and importantly, what approaches might be used to push cells toward a cochlear hair cell fate?
- 4) There were no functional assays presented. Other, recent characterizations of hair cells generated in vitro have demonstrated functional properties of using FM1-43 or electrophysiological assays. Might similar functional characterization be possible here?

Reviewer #2 (Remarks to the Author):

This manuscript from Roccio et al tackles a very interesting research topic in the auditory field. Currently, there is quite a wealthy amount of literature on murine models that have investigated the early, post-mitotic, stages of development of the auditory sensory organ. However, we know almost nothing about how these findings relate to human development, which is crucial for the future development of gene-based therapeutic strategies. In this paper, the authors have used a combination of complementary experimental approaches to provide a molecular characterization of the sensory cells and neuronal progenitors from the human fetal cochlea. Moreover, they have developed a method to culture post-mitotic hair cell progenitors, which show proliferating potential. The conclusions of the paper are at least in part (see comments below) substantiated by the results and the experiments are well executed. This is an important study and will add valuable data to the field. However, there are several aspects of the work that need some additional considerations and/or clarifications in order to substantially advance the field and as such stimulate future studies.

Major comments

- 1) The paper provides a molecular characterization of the newly nascent cells in the human cochlea. Although this is very useful, it mainly corroborates previous data on murine systems, and as such it is rather a confirmatory study more than taking the field forward. In order for this study to make a clear impact in the field, which is within the aims of this journal, the whole transcriptome analysis would be more appropriate and should be

provided. This will stimulate future work in the murine system, which is one of the main rationales for this study.

2) The claim about the presence of the "hair bundle" is extremely questionable. The fact that these protrusions express actin does not prove that they are stereocilia (hair bundle) – microvilli express actin and are not stereocilia. The authors should provide evidence that stereocilia-specific markers are expressed in these protrusions. Similarly, some caution should be used when stating that you have been able to generate hair cells (e.g. pg. 9) – you have not provided any functional studies and the expression of a few early marker genes does not make them hair cells.

3) Unfortunately the Discussion is very minimal and inadequate, which is very disappointing considering the quality of the data. There is very little or no attempt to integrate the present data with the published literature in a way that can be beneficial to the general readership of this journal. Previous work that has established different lineages of stem/progenitor cells from the human fetal cochlea are completely ignored. Moreover some of the claims are questionable and not explained. The statement that this study "demonstrates that hair cells were successfully generated in vitro from progenitors" is unsubstantiated. As explained above, there are not functional data in this work demonstrating the generation of hair cells. Finally, the last paragraph of the discussion is extremely speculative and the meaning is obscure; the authors have also provided several "impressive" statements without explaining "how" exactly this work will most likely move the field forward.

Minor comments

1) It would be useful to explain why the authors are only using Ki67 labelling to explore proliferation in vitro since the most commonly used assay is EdU or BrdU incorporation, which is a lot more informative.

2) It is not clear to me which controls have been made to ensure that the cells collected with FACS sorting represented a homogenous population of hair cell precursors.

3) The sample size for most of the experiments, especially the immuno, is not clear. This should be clearly stated.

4) What control procedures have been established to ensure that the fetal material obtained from donors was healthy and not already in an advance phase of degeneration?

5) There is a discrepancy in the age range used between the Abstract/Results (W8-12) and Method (W6-12).

6) Just a comment: wouldn't it be more "ethical" to call the donating persons "donors" instead of "mothers"?

Reviewer #3 (Remarks to the Author):

This manuscript from the laboratories of Roccio and Senn provides an immunologically based description of the cochlear duct from week 8 to week 12 human fetuses. It then follows on to successfully generate organoids from similar human cochlear cells. The study provides potentially valuable data in that it demonstrates some useful markers for identification and isolation of cochlear cells and then goes on to show that those cells can be expanded using organoid technology. However, I have concerns regarding the specificity of some of the antibodies used, in particular because no documentation is provided to confirm that the antibodies used work in humans. Further, the yield of hair cells from the organoids seems very modest by comparison with recent work from Kohler and Hashino using human ES or iPS cells. This, combined with the difficulty in obtaining the samples used here, especially by comparison with ES or iPS cells, raises questions as to the usefulness of the approach described here. Based on these findings, which largely represent a technical, rather than a conceptual, advance, this study might be better suited for a more specialized journal.

Specific concerns

As discussed, a significant concern is the ability of the antibodies used to bind to human protein epitopes. Details need to be provided on all the antibodies and some degree of validation, either, commercial validation, previous publication or new data to confirm the specificity of each antibody used in human tissue.

Figure 1f: LGR5 labeling is not convincing and MYO7A background seems very high

Figure 1g: I have concerns about bleed through or non-specific labeling between the LGR5 and JAG1 channels as the labeling looks very similar.

Figure 1h: while the W12 sections look very nice, the W11 images show very high levels of background. MYO7A appears to be expressed in the nucleus of every cell in each section.

Figure 4e': EPCAM and ECAD seem to be yielding the same labeling patterns. Could this be bleed through or non-specific labeling? Also, in 4e''', FBXO2, a transcription factor, is specifically excluded from cell nuclei, is this expected?

Reviewer #1 (Remarks to the Author):

Review of Roccio et al. Nature Communications

For this submission, Roccio and colleagues harvested inner ear tissue from aborted human fetuses between six and 12 weeks post-conception. The tissue was characterized using markers for the prosensory domain (Sox2), stem cell markers (LGR5), markers for cell cycle exit (p27Kip1) and markers that indicate emergence of hair cells (Myo7a). Use of these immuno markers has established the developmental sequence in the mouse cochlea, but how and whether these markers could be used to track human cochlear development was unknown. As such, the investigators established a critical period for human organ of Corti specification between week 8 and 12, which seems to parallel mouse inner development between, embryonic day 12 and 13.

The authors went on to use an qRT-PCR-based approach to characterize expression of 190 genes of particular interest in the inner ear and tracked expression at two developmental stages from three distinct regions. This analysis significantly extends the usefulness and impact of this work.

The authors also screened a series of cell surface markers in hopes of finding some useful for cell sorting. They identified EPCAM, which labelled cells of the developing cochlear duct. Sorted tissue yielded as many as 50,000 cells which were confirmed to be Lgr5-positive. The proliferative capacity of the cells was validated and there was confirmation using assays for additional markers. As expected, EPCAM- cells did not meet these criteria.

CD271 was used as a second marker to sort hair cell progenitors. Using double selection for both EPCAM and CD271 increased yield. In the second half of the manuscript, the authors used the EPCAM+ and /or CD271+ cells to generate inner ear organoids using what are now standard techniques. Hair cells numbers were low, but convincing.

The manuscript is well written and presented in logical easy to follow fashion. Figures are clearly labelled and the images and data are of excellent quality. In all these respects, the manuscript is consistent with the high standards of the Heller / Senn duo and importantly, this characteristic seems to be true of corresponding author, Marta Roccio, as well.

While the work is not hypothesis driven, it is a thorough first-of-its-kind of molecular and genetic characterization of human inner ear development, as well as, a demonstration that cells harvested from the fetal inner prosensory domain, can be used to generate human hair cells. The work will be an important point of reference for future studies aimed at reproducing human inner ear development to promote restoration of inner ear structure and function. As such, the manuscript will be of broad interest to inner ear biologists, as well as others interested in human development and regenerative medicine.

Below are listed a few additional points to consider that may further boost the impact of this work.

- 1) Despite some optimization, the numbers of hair cells produced was low relative to other mouse and human organoid protocols. For this approach to be broadly useful, some

further commentary on potential methods to boost yield would be helpful.

The yield is indeed limited at this moment. In the experiments where the double positive sorted population (EPCAM+/CD271+) was used, we obtained an average of 137 hair cell-like cells per organoid, and only two or 3 organoids are derived from the starting sorted population per human sample. Additional approaches to either immortalize or partially reprogram these cells, in order to derive readily expandable cell population, maintaining their characteristics of hair cell progenitors, could lead to better outcomes. These considerations have been included in the discussion.

2) To confirm cell identity, hair cell markers are useful, but of course, the key identifying characteristic of a hair cell is the hair bundle, which is only shown for one cell, in one image. Even with the low yield, might there be other hair bundle images that could be presented to boost confidence in the approach?

We have modified figure 6 and show more examples of MYO7A positive cells expressing F-Actin positive bundles that also show immunoreactivity with the Espin antibody. While many cells in the organoids display F-Actin positive bundles, the packed organization does not always allow for morphological assessment.

3) The one hair bundle shown appears to be vestibular in nature, but was presumably derived from cells harvested from the cochlear duct. This appears to be a recurrent theme in organoid and other protocols used to generate hair cells. Could the authors comment on why a vestibular hair cell phenotype seems to be the predominate hair cell fate, and importantly, what approaches might be used to push cells toward a cochlear hair cell fate?

The morphology of the bundle has not been carefully assessed in our study. As mentioned above, the packed organization in many instances of the MYO7A+ cells, displaying F-Actin+ bundles, and the limitation in imaging the sample from different direction (organoids are mounted on the insert membrane used for co-culture. In addition, to be able to use 63x objectives for bundle visualization we need to mount sample between coverglasses, causing in some cases squeezing of the sample) prevents proper morphological assessment.

Optimization of the culture conditions allowing for example better medium flow (now static) may improve the culture (viability/oxygen/nutrient perfusions) and as well as could improve the cellular organization and possibly bundle maturation. We have extended these points in the discussion.

4) There were no functional assays presented. Other, recent characterizations of hair cells generated in vitro have demonstrated functional properties of using FM1-43 or electrophysiological assays. Might similar functional characterization be possible here?

In absence of fluorescence reporters, generated by knock-in, it is not possible to visualize hair cells within the organoids, and therefore electrophysiology on these specimens is not feasible. We have performed experiments with FM1-43 loading. These have been now included in figure 6.

Reviewer #2 (Remarks to the Author):

This manuscript from Roccio et al tackles a very interesting research topic in the auditory field. Currently, there is quite a wealthy amount of literature on murine models that have investigated the early, post-mitotic, stages of development of the auditory sensory organ. However, we know almost nothing about how these findings relate to human development, which is crucial for the future development of gene-based therapeutic strategies. In this paper, the authors have used a combination of complementary experimental approaches to provide a molecular characterization of the sensory cells and neuronal progenitors from the human fetal cochlea. Moreover, they have developed a method to culture post-mitotic hair cell progenitors, which show proliferating potential. The conclusions of the paper are at least in part (see comments below) substantiated by the results and the experiments are well executed. This is an important study and will add valuable data to the field. However, there

are several aspects of the work that need some additional considerations and/or clarifications in order to substantially advance the field and as such stimulate future studies.

Major comments

1) The paper provides a molecular characterization of the newly nascent cells in the human cochlea. Although this is very useful, it mainly corroborates previous data on murine systems, and as such it is rather a confirmatory study more than taking the field forward. In order for this study to make a clear impact in the field, which is within the aims of this journal, the whole transcriptome analysis would be more appropriate and should be provided. This will stimulate future work in the murine system, which is one of the main rationales for this study.

We show here that many of the markers and processes described in mouse literature seem indeed to be conserved in human. This may be confirmative, but as the reviewer realizes, it is an important required validation for the field to move forward and translate findings into potential therapies.

Molecular therapy approaches for hearing loss for example are largely being developed based on the assumption that the same signaling pathways present in mouse would be present and targetable in humans, but evidence is limited or absent.

The *in vitro* culture of sorted human progenitors/cochlear duct resident cells and derivation of culture conditions that allows for hair cell differentiation is novel and unique.

We are well aware that the transcriptional profile of the sorted inner ear progenitors at different developmental stages, and possibly at the single cell level would be an extremely useful data set for the community. Unfortunately it goes beyond our current possibility.

This is not a trivial experiment, given the fact that the tissue is very rare and the collection very slow. Gathering of biological “replicates” for each developmental stage to provide solid evidence for temporal regulation will require additional years.

(The samples used for this study have been collected within the time span of 4 years)

Such a data set will be an independent study. Not having this data at the moment does not change any of the conclusions of the current manuscript.

2) The claim about the presence of the “hair bundle” is extremely questionable. The fact

that these protrusions express actin does not prove that they are stereocilia (hair bundle) – microvilli express actin and are not stereocilia. The authors should provide evidence that stereocilia-specific markers are expressed in these protrusions. Similarly, some caution should be used when stating that you have been able to generate hair cells (e.g. pg. 9) – you have not provided any functional studies and the expression of a few early marker genes does not make them hair cells.

We have performed additional experiments and show the presence of F-Actin+ bundles that also display immunoreactivity with the Espin antibody. We have moreover included FM1-43 staining experiments in samples that were kept in differentiation for 4 weeks. We refer to these cells as hair cell-like cells.

3) Unfortunately the Discussion is very minimal and inadequate, which is very disappointing considering the quality of the data. There is very little or no attempt to integrate the present data with the published literature in a way that can be beneficial to the general readership of this journal. Previous work that has established different lineages of stem/progenitor cells from the human fetal cochlea are completely ignored. Moreover some of the claims are questionable and not explained. The statement that this study “demonstrates that hair cells were successfully generated in vitro from progenitors” is unsubstantiated. As explained above, there are not functional data in this work demonstrating the generation of hair cells. Finally, the last paragraph of the discussion is extremely speculative and the meaning is obscure; the authors have also provided several “impressive” statements without explaining “how” exactly this work will most likely move the field forward.

We are aware of the brief discussion. This was the consequence of the word limits imposed by the editorial policy of the journal where we initially submitted our work. We have now extended the discussion.

Minor comments

1) It would be useful to explain why the authors are only using Ki67 labelling to explore proliferation in vitro since the most commonly used assay is EdU or BrdU incorporation, which is a lot more informative.

Edu is more informative when one wants to analyze if the cells have transited through S phase at specific time points during the culture.

Short incubation would give a snapshot picture of how many cells are in S-phase in that defined time window, which is equivalent to snap shots of proliferative cells assessed by KI67 staining. The latter including more cells, as all cell cycle phases are taken into account.

As the organoids expands in culture quite robustly, Edu/BrdU addition in culture, for the entire time, would likely reveal proliferation. However we have observed toxic effect when cells are exposed for prolonged time.

2) It is not clear to me which controls have been made to ensure that the cells collected with FACS sorting represented a homogenous population of hair cell precursors.

FACS sorted cells are checked by purity, however these populations are not homogeneous.

The cells collected with EPCAM only sorting, do not represent a homogeneous population. They include all cochlear duct resident cell types.

That's why, in this scenario the differentiation efficiency is rather low: Committed precursors from stria vascularis, and Reissner's membrane for example, resident in the CD will not differentiate into hair cells.

The double positive sorted population (EPCAM+/CD271+) pool isolated by FACS is also not "homogeneous" and the number and types of cells that are included varies with developmental stages. Different cells in the PSD express this marker, as seen by immunostaining of the section.

Possibly, the younger the sample, the less committed the cells to a defined fate, the more "homogeneous" the sorted pool may be. Namely all cells could still be able to revert fate from supporting cell to hair cell.

At later stages, when supporting cells may have gained their identity, the specific differences in the sorted cells belonging to the double positive pool may be a strong determinant of cell heterogeneity. We have better addressed these points in the discussion

3) The sample size for most of the experiments, especially the immuno, is not clear. This should be clearly stated.

The samples used are indicated in the supplementary table. Samples are grouped by developmental week and per experiment/experiment type.

For the staining characterization by histology 3 samples/stage are used. For W12 samples n=2. We have now indicated this better in the methods.

4) What control procedures have been established to ensure that the fetal material obtained from donors was healthy and not already in an advanced phase of degeneration?

We are usually informed the previous day about an abortion procedure with a signed informed consent that could be included in the study. We collect the sample the same day as the abortion is performed as soon as notified. There's however variations in the elapsed time to tissue collection. If the tissue is not collected the same day, we discard it. Broken inner ears (with visible blood inclusion) are not used. We don't have additional parameters for testing degeneration of the sample.

Notably, in parallel to the collection of the inner ear, we also harvested brain tissue for tissue cultures. The latter demonstrated a good survival of neurons over a culture period of one week indicating that the fetal tissue was not in an advanced stage of degeneration.

5) There is a discrepancy in the age range used between the Abstract/Results (W8-12) and Method (W6-12).

We have indeed analyzed younger samples. Some were used for staining controls that do not appear in the paper. We have left it unchanged as the characterization mainly refers to the stages W8-12.

6) Just a comment: wouldn't it be more "ethical" to call the donating persons "donors" instead of "mothers"?

We have corrected this.

Reviewer #3 (Remarks to the Author):

This manuscript from the laboratories of Roccio and Senn provides an immunologically based description of the cochlear duct from week 8 to week 12 human fetuses. It then follows on to successfully generate organoids from similar human cochlear cells. The study provides potentially valuable data in that it demonstrates some useful markers for identification and isolation of cochlear cells and then goes on to show that those cells can be expanded using organoid technology. However, I have concerns regarding the specificity of some of the antibodies used, in particular because no documentation is provided to confirm that the antibodies used work in humans. Further, the yield of hair cells from the organoids seems very modest by comparison with recent work from Kohler and Hashino using human ES or iPS cells. This, combined with the difficulty in obtaining the samples used here, especially by comparison with ES or iPS cells, raises questions as to the usefulness of the approach described here. Based on these findings, which largely represent a technical, rather than a conceptual, advance, this study might be better suited for a more specialized journal.

The study describes for the first time the early stages of human cochlear development, which is per se new. Validation of the expression of well-characterized markers from mouse development in human specimens is an important aspect of translational research. The generation of human hair cells from ESC and iPSC offers great advantages for *in vitro* screening purposes. We are not questioning these aspects, nor proposing that the organoids generated by our approach would be more suitable for screening, neither in terms of efficiency nor degree of maturation than the pluripotent cells derived ones. We have implemented the discussion to address these concerns.

Specific concerns

As discussed, a significant concern is the ability of the antibodies used to bind to human protein epitopes. Details need to be provided on all the antibodies and some degree of validation, either, commercial validation, previous publication or new data to confirm the specificity of each antibody used in human tissue.

Figure 1f: LGR5 labeling is not convincing and MYO7A background seems very high
Figure 1h: while the W12 sections look very nice, the W11 images show very high levels of background. MYO7A appears to be expressed in the nucleus of every cell in each section

The antibody used in the study are listed in table 3

For the markers that we found to be negative, both on section staining as well as by flow-cytometry on fresh or fixed tissue (CD146, CD133 and CD15), the antibodies are well known antibodies used in human studies and known to work on human tissue. We believe the staining is negative.

CD271, CD326, and LGR5 are raised against the human epitope specifically.
CD49f and ECAD are reported to cross-react with human.

We agree with the reviewer LGR5 staining is not convincing, this was already mentioned in the text. The antibody is nevertheless human specific, raised in the Medema and Clevers lab (Kemper et al Stem Cells 2012), sold by BD. Also in the producer's hand the

expression at histological level is not very convincing. And similar data are shown for flow-cytometry, where the antibody detects well the overexpressed LGR5, but not the endogenous protein. This is a known issue in the field as far as I am aware of.

The antibody does not cross-react with mouse tissue (Lgr5-GFP), which we used initially to assess the staining, nor stains human fetal intestinal crypts very clearly (data not shown).

As the idea that Lgr5 expressing cells act as hair cell progenitors in the young postnatal cochlea is being pushed forward in the literature, we believe it is of interest to show the data. Emphasis has now been placed in the text on the faint expression/detection issue.

We have included new data where LGR5 expression was tested using RNAscope technology showing the expression of LGR5 mRNA in the PSD. The expression is robust and quite comparable to the levels detected in the intestinal crypts of fetal sample of the same age.

Concerning MYO7A staining, the background staining is visible only in sample where hair cells have not appeared yet. This is done on purpose to display that there's no morphologically distinct cell type expressing MYO7A in the cytoplasm before W11.

In addition, the panels in figure 1 (W11 and W12) have been imaged simultaneously in order to prepare the figure for the manuscript, but immunostained at different time points. Specifically, the W11 sample is a much older sample (2 years older), and the signal may have faded. I here provide the original image for the basal turn acquired on a fresh sample (W11 basal turn 63x).

The reviewer should realize that despite not ideal, the long time intervals between the collection of each human sample does not always allow for sample storage without proceeding with analysis.

Figure 1g: I have concerns about bleed through or non-specific labeling between the LGR5 and JAG1 channels as the labeling looks very similar.

All images are acquired in sequential mode on a LSM confocal.

Similar results were obtained also in sections where the two stainings were not combined.

As mentioned above, the Lgr5 staining is very weak.

JAG1 staining seems to be higher in the region surrounding the PSD, however the signal is weak and the localization does not entirely match the mouse literature.

We have used two commercially available JAG1 antibody, both reported to work in human tissue, for validation. In both cases the staining is faint.

We have removed the panel from figure 1 as we feel we are not able to provide better evidence that the weak staining is the results of staining limitations or lower expression compared to the murine PSD. The text has been modified accordingly

Figure 4e': EPCAM and ECAD seem to be yielding the same labeling patterns. Could this be bleed through or non-specific labeling?

All images are acquired in sequential mode on a LSM confocal (single laser/PMT active). The staining for ECAD and EPCAM in the organoids culture are performed using the same antibodies as in the sections (see supplementary figure 3). Here ECAD staining seems to have much higher background staining than EPCAM, outside of the epithelial cells of the cochlear duct.

We provide for the reviewer an additional figure showing organoids stained with both or single markers and imaged under the exact same conditions for all channels to show that there's no aspecific bleed through.

*Organoids are stained with single antibody or double stained.
Imaging is performed for all channels with the same parameters.*

Also, in $4e''$, FBXO2, a transcription factor, is specifically excluded from cell nuclei, is this expected?

FBXO2 is not a transcription factor. It's a E3 ligase. The immunoreactivity reported by Hartman et al. *Frontiers in cellular neuroscience* 2015 in the cochlear duct, also shows exclusion from the nucleus. Similar results were obtained by Kohler and Hashino (*Nat Biotech* 2017) with human organoids, where FBXO2 staining is also excluded from the nucleus (see supplementary material).

We provide here a panel showing the FBXO2 immunoreactivity in cochlear ducts of two different stages of development (W6 and W11), stained with 3 commercially available antibodies for FBXO2. In all cases the staining is consistent and not-nuclear, and appears comparable to what we have obtained in the organoids staining.

FBXO2 (SC A12)
a-mouse-555

FBXO2 (SC goat)
a-goat-488

FBXO2 (AbCAM)
a-rabbit-555

FBXO2 (AbCAM)
a-rabbit-555

W5.8

E12.24

W11

FBXO2 (W5.8)

Reviewers' comments:

Reviewer #1 (Remarks to the Author):

The authors have satisfied my requests and are to be congratulated on a lovely body of work.

Reviewer #2 (Remarks to the Author):

Although the authors have addressed one of my major concerns, there are still some unresolved issues. The conclusion referring to the presence of the "hair bundle" or even functional hair cell-like cells is still unsubstantiated. First of all, the new immunostaining experiments with Espin are impossible to assess; espin seems to be expressed everywhere and as such the labelling appears to be very unspecific. Additionally, the claim that FM1-43 uptake by these cells is a read-out of hair cell/bundle functionality is wrong. FM1-43 can also enter cells via endocytosis, even when used for short duration. As a minimal step towards proving that mechano-electrical transduction (MT) channels are present, FM1-43 uptake should be prevented by using an MT blocker. The statement in the discussion (pg. 304-306) suggesting that the generation of fluorescent reporter lines is required to probe the mechanosensitivity properties of the hair bundle is also wrong; it only requires a mechanical displacement of the presumptive hair-like bundle.

The second problem I have is still related to the discussion, which has been extended but still provides very little insights into how this work will be able to move the field forward. Most of the previous work that has established different lineages of stem/progenitor cells from the human fetal cochlea is still missing or poorly described. I am left puzzled about the remit of this work. Is this simply a methods paper? or does it represent a fundamental breakthrough? If the latter is true, why is that? This is not clear from the very general (review-type) discussion.

Minor comments

If W6 samples have not been included in the paper, then reference to these data should be removed (state W8-12 in the Methods).

Reviewer #3 (Remarks to the Author):

The authors have done a reasonable job of addressing the reviewer's comments. With regard to the newly added FM1-43X experiments, the density of cells that are positive for uptake of FM1-43X seems higher by comparison with expression of MYO7A. Was a comparison made of the two? Not necessarily in the same sample, but just in terms of estimates of overall hair cell density?

Also, in the rebuttal to Reviewer 2, ESPIN is indicated as a stereocilia marker. This is not

correct in the context of the Reviewer's concern. ESPIN is also expressed in other types of microvilli in addition to stereocilia.

Figure 6i: MYO7A also localizes to stereocilia, therefore, how was it possible to discriminate ESPIN from MYO7A if both were done using the same fluorophore?

Beyond these comments, my initial concern, that the conceptual advance is limited here remains. Yes, it is important from a translational sense to confirm that mouse and human processes are conserved, but does that alone make a study appropriate for this journal? I am not convinced of that, but in the end that is a decision best suited for the editors.

We would like to thank the reviewers for the comments and criticisms that have substantially improved the quality of the manuscript.

Below a point to point reply

Reviewers' comments:

Reviewer #1 (Remarks to the Author):

The authors have satisfied my requests and are to be congratulated on a lovely body of work.

Reviewer #2 (Remarks to the Author):

Although the authors have addressed one of my major concerns, there are still some unresolved issues. The conclusion referring to the presence of the “hair bundle” or even functional hair cell-like cells is still unsubstantiated. First of all, the new immunostaining experiments with Espin are impossible to assess; espin seems to be expressed everywhere and as such the labelling appears to be very unspecific. Additionally, the claim that FM1-43 uptake by these cells is a read-out of hair cell/bundle functionality is wrong. FM1-43 can also enter cells via endocytosis, even when used for short duration. As a minimal step towards proving that mechanoelectrical transduction (MT) channels are present, FM1-43 uptake should be prevented by using an MT blocker. The statement in the discussion (pg. 304-306) suggesting that the generation of fluorescent reporter lines is required to probe the mechanosensitivity properties of the hair bundle is also wrong; it only requires a mechanical displacement of the presumptive hair-like bundle.

We provide now novel images showing ESPIN staining in cells that have been co-labeled with BRN3C and also were tested for FM1-43 uptake (Fig. 6 and Supplementary Fig.8 and annex A). We have replaced the previous images as these raised concerns. The use of the same fluorophore for Espin and Myosin7a in the previous version was dictated by the fact that the only two reported antibodies validated on human samples for these markers were of rabbit origin. The samples were nevertheless sequentially stained and imaged for the two markers, which allowed us to conduct these experiments. The experimental procedure was not described in detail in the methods but mentioned in the figure legend (annex B).

We have now added new data obtained with two independent samples (W9.6 and W12), showing the expression of ESPIN in apical bundles. The cells were co-labeled with a BRN3C antibody, strongly expressed in the nucleus of the cells. The antibodies were validated in human vestibular tissue (new supplementary figure2) as well as on W12 human cochlea sections. (annex C).

Concerning the FM1-43 uptake, the reviewer is right in mentioning aspecific uptake.

We have observed FM1-43 labeling in cells with possibly leaky membranes (apoptotic), which appeared to have small sizes and bright loading. These are present in samples that are either imaged “live” or imaged immediately after fixation prior to staining, but then eliminated with consecutive washing steps during immunostaining.

We have now performed the FM1-43 loading (with a fixable version of the dye: FM1-43 FX) in two different fetal samples, combined this with immunostaining and inhibitor treatment.

Given the small number of organoids obtained from the W9.6 sample, hair cell-like cells were exclusively tested for dye uptake in presence of (and 10'pre-treatment with) with a general inhibitor of endocytosis (concanavalin A). FM1-43 was added for 30 seconds at the concentration of 5 μ M, and quickly washed off with cold PBS.

The images provided (Fig 6 i-k) show very bright accumulation of the dye at the bundle level, counterstained for ESPIN, as well and in the cytoplasm. We have performed similar experiment in parallel with rodent organ of Corti explants and obtained comparable results (data not shown). With the W12 sample we have been able to quantify FM1-43 uptake in samples that were treated with Concanavalin A or Curare (at 1mM). MYO7A positive cells from 2-3 different organoids were quantified and indeed we detected a decreased fluorescence intensity in samples incubated with Curare, previously shown to acting as MET channel competitive blocker ⁽¹⁾.

We have also tested the uptake of Gentamycin-Texas Red in one of the samples (W9.6). Hair cell-like cells with a more mature morphology and actin bundles displayed GTTR uptake, while a number of other cells in the organoid showing MYO7A immunoreactivity but absence of bundles did not.

We believe the two experiments suggest functional maturation of the *in vitro* derived hair cell-like cells.

The generation of fluorescent reporters had been indicated in the discussion as a pre-requisite for identification of the cells to be patched, not for activity recording per se. We are not able using this type of 3D

culture, to assess the location of the generated hair cells by light microscopy, neither to see their bundles to perform patch clamp/bundle displacement as the reviewer suggests. We have now clarified the statement. We have also consulted with a hair cell physiologist (Anthony Ricci, Stanford University) who explained to us that he would need a number of samples to become acquainted with the morphology of the preparation even if the hair cell-like cells would be GFP-labeled before even attempting a recording. I hope the reviewer will understand the little fetal material available currently does not allow for this type of experiments.

The second problem I have is still related to the discussion, which has been extended but still provides very little insights into how this work will be able to move the field forward. Most of the previous work that has established different lineages of stem/progenitor cells from the human fetal cochlea is still missing or poorly described. I am left puzzled about the remit of this work. Is this simply a methods paper? or does it represent a fundamental breakthrough? If the latter is true, why is that? This is not clear from the very general (review-type) discussion.

We have modified in part the manuscript discussion

Minor comments

If W6 samples have not been included in the paper, then reference to these data should be removed (state W8-12 in the Methods).

We have modified the text

Reviewer #3 (Remarks to the Author):

The authors have done a reasonable job of addressing the reviewer's comments. With regard to the newly added FM1-43X experiments, the density of cells that are positive for uptake of FM1-43X seems higher by comparison with expression of MYO7A. Was a comparison made of the two? Not necessarily in the same sample, but just in terms of estimates of overall hair cell density?

We have replaced the images concerning the uptake of the FM1-43 dye and performed additional experiments assessing uptake in presence of inhibitors (endocytosis (concanavalin A) and MET channel (Curare)). We have also now performed immunostaining in order to assess the uptake of FM1-43 specifically in MYO7A+ cells. These data are presented in Figure 6 (i-l) as well as in supplementary figure 9. We believe they provide good evidence for the MET channel mediated FM1-43 uptake.

Aspecific staining with the FM1-43 dye has been observed in cells that appear dead or dying (small/fragmented nuclei) probably because of leaky membranes. These are further removed from the preparation with washing steps during the immunostaining of the samples. This was the case for some of the cells in the images presented in the previous version.

Also, in the rebuttal to Reviewer 2, ESPIN is indicated as a stereocilia marker. This is not correct in the context of the Reviewer's concern. ESPIN is also expressed in other types of microvilli in addition to stereocilia.

We have modified the text and provided novel images for Espin in figure 6 and supplementary figure 8.

Figure 6i: MYO7A also localizes to stereocilia, therefore, how was it possible to discriminate ESPIN from MYO7A if both were done using the same fluorophore?

The reason for the use of the same fluorophore for Espin and Myosin7a was dictated by the fact that the only two reported antibodies validated on human samples for these markers were of rabbit origin. The samples were nevertheless sequentially stained and imaged for the two markers, which allowed us to conduct these experiments. The experimental procedure was not described in detail in the methods but mentioned in the figure legend.

We have now replaced the image as it raised concerns to both reviewer #2 and #3 and provide novel images showing ESPIN positivity in hair bundles.

We validated the used of BRN3C antibody (mouse monoclonal) for staining the nucleus of hair cells both in the vestibular organ (supplementary figure 2 new) and in the cochlea. We could identify several BRN3C+/ESPIN+ hair cells displaying FM1-43 uptake.

Beyond these comments, my initial concern, that the conceptual advance is limited here remains. Yes, it is important from a translational sense to confirm that mouse and human processes are conserved, but does that alone make a study appropriate for this journal? I am not convinced of that, but in the end that is a decision best suited for the editors.

1. Alharazneh, A., *et al.* Functional hair cell mechanotransducer channels are required for aminoglycoside ototoxicity. *PLoS One* **6**, e22347 (2011).

Annex A

Specificity of the fluorescent signal was assessed in hair cell-like cells immunostained for ESPIN (Alexa 647) and BRN3C (Alexa-488) after being exposed to FM1-43 (upper panel) or left untreated (lower). Images were acquired with the same parameters (laser intensity/gain/filters) for the two images simultaneously.

Annex A

Annex B

Sequential imaging and staining for MYO7A and ESPIN.

Organoids were first immunostained for MYO7A and imaged by confocal microscopy. The same sample was then re-stained for ESPIN and F-Actin and re-imaged.

MYO7A and ESPIN (both rabbit polyclonal) were detected with an anti-rabbit-Alexa-555 antibody

Annex B

staining1:
MYO7A-Alexa fluo 555

staining2:
ESPIN-Alexa fluo 555
phalloidin-ATTO488

Annex C

Immunostaining on human fetal cochlea (W12) with BRN3C antibody show specific localization into nascent hair cells

Annex C

Human fetal cochlea W12

REVIEWERS' COMMENTS:

Reviewer #2 (Remarks to the Author):

In the revised version of the ms the Authors have fully addressed my major points.